# FedImpro: Measuring and Improving Client Update in Federated Learning

**Zhenheng Tang**[1,*]   **Yonggang Zhang**[1]   **Shaohuai Shi**[2]   **Xinmei Tian**[3]
**Tongliang Liu**[4]   **Bo Han**[1]   **Xiaowen Chu**[5,†]
[1] Department of Computer Science, Hong Kong Baptist University
[2] Harbin Institute of Technology, Shenzhen
[3] University of Science and Technology of China [4] Sydney AI Centre, The University of Sydney
[5] DSA Thrust, The Hong Kong University of Science and Technology (Guangzhou)

## Abstract

Federated Learning (FL) models often experience client drift caused by heterogeneous data, where the distribution of data differs across clients. To address this issue, advanced research primarily focuses on manipulating the existing gradients to achieve more consistent client models. In this paper, we present an alternative perspective on client drift and aim to mitigate it by generating improved local models. First, we analyze the generalization contribution of local training and conclude that this generalization contribution is bounded by the conditional Wasserstein distance between the data distribution of different clients. Then, we propose `FedImpro`, to construct similar conditional distributions for local training. Specifically, `FedImpro` decouples the model into high-level and low-level components, and trains the high-level portion on reconstructed feature distributions. This approach enhances the generalization contribution and reduces the dissimilarity of gradients in FL. Experimental results show that `FedImpro` can help FL defend against data heterogeneity and enhance the generalization performance of the model.

## 1 Introduction

The convergence rate and the generalization performance of FL suffers from heterogeneous data distributions across clients (Non-IID data) (Kairouz et al., 2019). The FL community theoretically and empirically found that the "client drift" caused by the heterogeneous data is the main reason of such a performance drop (Guo et al.; Wang et al., 2020a). The client drift means the far distance between local models on clients after being trained on private datasets.

Recent convergence analysis (Reddi et al., 2021; Woodworth et al., 2020) of FedAvg shows that the degree of client drift is linearly upper bounded by gradient dissimilarity. Therefore, most existing works (Karimireddy et al., 2020; Wang et al., 2020a) focus on gradient correction techniques to accelerate the convergence rate of local training. However, these techniques rely on manipulating gradients and updates to obtain more similar gradients (Woodworth et al., 2020; Wang et al., 2020a; Sun et al., 2023a). However, the empirical results of these methods show that there still exists a performance gap between FL and centralized training.

In this paper, we provide a novel view to correct gradients and updates. Specifically, we formulate the objective of local training in FL systems as a generalization contribution problem. The generalization contribution means how much local training on one client can improve the generalization performance on other clients' distributions for server models. We evaluate the generalization performance of a local model on other clients' data distributions. Our theoretical analysis shows that the generalization contribution of local training is bounded by the conditional Wasserstein distance between clients' distributions. This implies that even if the marginal distributions on different clients are the same, it is insufficient to achieve a guaranteed generalization performance of local training. Therefore, the key

---

[†] Corresponding author (xwchu@ust.hk).
[*] This work is partially done during the internship in The Hong Kong University of Science and Technology (Guangzhou).

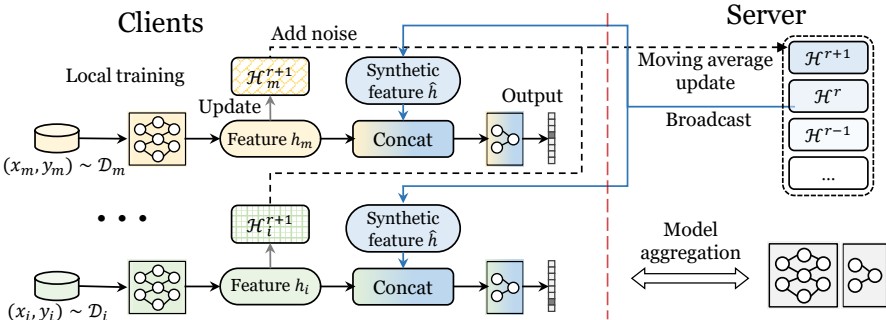

Figure 1: Training process of our framework. On any $m$-th Client, the low-level model uses the raw data $x_m$ as input, and outputs feature $h_m$. The high-level model uses $h_m$ and samples $\hat{h}$ from a shared distribution $\mathcal{H}^r$ as input for forward and backward propagation. Noises will be added to the locally estimated $\mathcal{H}_m^{r+1}$ before aggregation on the Server to update the global $\mathcal{H}^{r+1}$. Model parameters follow the FedAvg aggregation or other FL aggregation algorithms.

to promoting generalization contribution is to leverage the same or similar conditional distributions for local training.

However, collecting data to construct identical distributions shared across clients is forbidden due to privacy concerns. To avoid privacy leakage, we propose decoupling a deep neural network into a low-level model and a high-level one, i.e., a feature extractor network and a classifier network. Consequently, we can construct a shared identical distribution in the feature space. Namely, on each client, we coarsely[1] estimate the feature distribution obtained by the low-level network and send the noised estimated distribution to the server model. After aggregating the received distributions, the server broadcasts the aggregated distribution and the server model to clients simultaneously. Theoretically, we show that introducing such a simple decoupling strategy promotes the generalization contribution and alleviates gradient dissimilarity. Our extensive experimental results demonstrate the effectiveness of FedImpro, where we consider the global test accuracy of four datasets under various FL settings following previous works (He et al., 2020b; Li et al., 2020b; Wang et al., 2020a).

Our main contributions include: (1) We theoretically show that the critical, yet far overlooked generalization contribution of local training is bounded by the conditional Wasserstein distance between clients' distributions (Section 4.1). (2) We are the first to theoretically propose that sharing similar features between clients can improve the generalization contribution from local training, and significantly reduce the gradient dissimilarity, revealing a new perspective of rectifying client drift (Section 4.2). (3) We propose FedImpro to efficiently estimate feature distributions with privacy protection, and train the high-level model on more similar features. Although the distribution estimation is approximate, the generalization contribution of the high-level model is improved and gradient dissimilarity is reduced (Section 4.3). (4) We conduct extensive experiments to validate gradient dissimilarity reduction and benefits on generalization performance of FedImpro (Section 5).

## 2 RELATED WORKS

We review FL algorithms aiming to address the Non-IID problem and introduce other works related to measuring client contribution and decoupled training. Due to limited space, we leave a more detailed discussion of the literature review in Appendix D.

### 2.1 ADDRESSING NON-IID PROBLEM IN FL

**Model Regularization** focuses on calibrating the local models to restrict them not to be excessively far away from the server model. A number of works like FedProx (Li et al., 2020b), FedDyn (Acar et al., 2021), SCAFFOLD (Karimireddy et al., 2020). VHL (Tang et al., 2022b) utilizes shared noise data to calibrate feature distributions. FedETF (Li et al., 2023b) proposed a synthetic and fixed ETF classified to resolve the classifier delemma. SphereFed (Dong et al., 2022) constrains the learned representations to be a unit hypersphere.

---

[1]Considering the computation and communication costs, avoiding privacy leakage, the distribution estimation is approximate. And the parameters will be sent out with noise.

**Reducing Gradient Variance** tries to correct the directions of local updates at clients via other gradient information. This kind of method aims to accelerate and stabilize the convergence, like FedNova (Wang et al., 2020a), FedOPT (Reddi et al., 2021) and FedSpeed (Sun et al., 2023b). Our theorem 4.2 provides a new angle to reduce gradient variance.

**Sharing data** methods proposes sharing the logits or data between clients (Yang et al., 2023). Cronus (Chang et al., 2019) shares the logits to defend the poisoning attack. CCVR (Luo et al., 2021) transmit the logits statistics of data samples to calibrate the last layer of Federated models. CCVR (Luo et al., 2021) also share the parameters of local feature distribution. FedDF (Lin et al., 2020) finetunes on the aggregated model via the knowledge distillation on shared publica data. The supriority of FedImpro may result from the difference between these works. Specifically, FedImpro focuses on the training process and reducing gradient dissimilarity of the high-level layers. In contrast, CCVR is a post-hoc method and calibrates merely the classifiers. Furthermore, the FedDF fails to distill knowledge when using the random noise datasets as shown in the results.

## 2.2 Measuring Contribution from Clients

Clients' willingness to participate in FL training depends on the rewards offered. Hence, it is crucial to evaluate their contributions to the model's performance (Yu et al., 2020; Ng et al., 2020; Liu et al., 2022; Sim et al., 2020). Some studies (Yuan et al., 2022) propose experimental methods to measure performance gaps from unseen client distributions. Data shapley (Ghorbani & Zou, 2019; Sim et al., 2020; Liu et al., 2022) is introduced to assess the generalization performance improvement resulting from client participation. These approaches evaluate the generalization performance with or without certain clients engaging in the entire FL process. However, we hope to understand the contribution of clients at each communication round. Consequently, our theoretical conclusions guide a modification on feature distributions, improving the generalization performance of the trained model.

## 2.3 Split Training

Some works propose Split FL (SFL) to utilize split training to accelerate federated learning (Oh et al., 2022; Thapa et al., 2020). In SFL, the model is split into client-side and server-side parts. At each communication round, the client only downloads the client-side model from the server, and conducts forward propagation, and sends the hidden features to the server to compute the loss and conduct backward propagation. These methods aim to accelerate the training speed of FL on the client side and cannot support local updates. In addition, sending all raw features could introduce a high risk of data leakage. Thus, we omit the comparisons to these methods.

## 2.4 Privacy Concerns

There are many other works (Luo et al., 2021; Li & Wang, 2019; He et al., 2020a; Liang et al., 2020; Thapa et al., 2020; Oh et al., 2022) that propose to share the hidden features to the server or other clients. Different from them, our decoupling strategy shares the parameters of the estimated feature distributions instead of the raw features, avoiding privacy leakage. We show that FedImpro successfully protect the original data privacy in Appendix F.7.

## 3 Preliminaries

### 3.1 Problem Definition

Suppose we have a set of clients $\mathcal{M} = \{1, 2, \cdots, M\}$ with $M$ being the total number of participating clients. FL aims to make these clients with their own data distribution $\mathcal{D}_m$ cooperatively learn a machine learning model parameterized as $\theta \in \mathbb{R}^d$. Suppose there are $C$ classes in all datasets $\cup_{m \in \mathcal{M}} \mathcal{D}_m$ indexed by $[C]$. A sample in $\mathcal{D}_m$ is denoted by $(x, y) \in \mathcal{X} \times [C]$, where $x$ is a model input in the space $\mathcal{X}$ and $y$ is its corresponding label. The model is denoted by $\rho(\theta; x) : \mathcal{X} \to \mathbb{R}^C$. Formally, the global optimization problem of FL can be formulated as (McMahan et al., 2017):

$$\min_{\theta \in \mathbb{R}^d} F(\theta) := \sum_{m=1}^{M} p_m F_m(\theta) = \sum_{m=1}^{M} p_m \mathbb{E}_{(x,y) \sim \mathcal{D}_m} f(\theta; x, y), \tag{1}$$

where $F_m(\theta) = \mathbb{E}_{(x,y)\sim\mathcal{D}_m} f(\theta;x,y)$ is the local objective function of client $m$ with $f(\theta;x,y) = CE(\rho(\theta;x),y)$, $CE$ denotes the cross-entropy loss, $p_m > 0$ and $\sum_{m=1}^{M} p_m = 1$. Usually, $p_m$ is set as $\frac{n_m}{N}$, where $n_m$ denotes the number of samples on client $m$ and $N = \sum_{m=1}^{M} n_m$.

The clients usually have a low communication bandwidth, causing extremely long training time. To address this issue, the classical FL algorithm FedAvg (McMahan et al., 2017) proposes to utilize local updates. Specifically, at each round $r$, the server sends the global model $\theta^{r-1}$ to a subset of clients $\mathcal{S}^r \subseteq \mathcal{M}$ which are randomly chosen. Then, all selected clients conduct some iterations of updates to obtain new client models $\{\theta_m^r\}$, which are sent back to the server. Finally, the server averages local models according to the dataset size of clients to obtain a new global model $\theta^r$.

## 3.2 GENERALIZATION QUANTIFICATION

Besides defining the metric for the training procedure, we also introduce a metric for the testing phase. Specifically, we define criteria for measuring the generalization performance for a given deep model. Built upon the margin theory (Koltchinskii & Panchenko, 2002; Elsayed et al., 2018), for a given model $\rho(\theta;\cdot)$ parameterized with $\theta$, we use the worst-case margin [2] to measure the generalizability on the data distribution $\mathcal{D}$:

**Definition 3.1.** (Worst-case margin.) Given a distribution $\mathcal{D}$, the worst-case margin of model $\rho(\theta;\cdot)$ is defined as $W_d(\rho(\theta),\mathcal{D}) = \mathbb{E}_{(x,y)\sim\mathcal{D}} \inf_{\arg\max_i \rho(\theta;x')_i \neq y} d(x',x)$ with $d$ being a specific distance, where the $\arg\max_i \rho(\theta;x')_i \neq y$ means the $\rho(\theta;x')$ mis-classifies the $x'$.

This definition measures the expected largest distance between the data $x$ with label $y$ and the data $x'$ that is mis-classified by the model $\rho$. Thus, smaller margin means higher possibility to mis-classify the data $x$. Thus, we can leverage the defined worst-case margin to quantify the generalization performance for a given model $\rho$ and a data distribution $\mathcal{D}$ under a specific distance. Moreover, the defined margin is always not less than zero. It is clear that if the margin is equal to zero, the model mis-classifies almost all samples of the given distribution.

## 4 DECOUPLED TRAINING AGAINST DATA HETEROGENEITY

This section formulates the generalization contribution in FL systems and decoupling gradient dissimilarity.

### 4.1 GENERALIZATION CONTRIBUTION

Although Eq. 1 quantifies the performance of model $\rho$ with parameter $\theta$, it focuses more on the training distribution. In FL, we cooperatively train machine learning models because of a belief that introducing more clients `seems` to contribute to the performance of the server models. Given client $m$, we quantify the "belief", i.e., the generalization contribution, in FL systems as follows:

$$\mathbb{E}_{\Delta:\mathbf{L}(\mathcal{D}_m)} W_d(\rho(\theta+\Delta), \mathcal{D}\backslash\mathcal{D}_m), \tag{2}$$

where $\Delta$ is a pseudo gradient [3] obtained by applying a learning algorithm $\mathbf{L}(\cdot)$ to a distribution $\mathcal{D}_m$, $W_d$ is the quantification of generalization, and $\mathcal{D}\backslash\mathcal{D}_m$ means the data distribution of all clients except for client $m$. Eq. 2 depicts the contribution of client $m$ to generalization ability. Intuitively, we prefer the client where the generalization contribution can be lower bounded.

**Definition 4.1.** The Conditional Wasserstein distance $C_d(\mathcal{D},\mathcal{D}')$ between the distribution $\mathcal{D}$ and $\mathcal{D}'$:

$$C_d(\mathcal{D},\mathcal{D}') = \frac{1}{2}\mathbb{E}_{(\cdot,y)\sim\mathcal{D}} \inf_{J\in\mathcal{J}(\mathcal{D}|y,\mathcal{D}'|y)} \mathbb{E}_{(x,x')\sim J} d(x,x') + \frac{1}{2}\mathbb{E}_{(\cdot,y)\sim\mathcal{D}'} \inf_{J\in\mathcal{J}(\mathcal{D}|y,\mathcal{D}'|y)} \mathbb{E}_{(x,x')\sim J} d(x,x').$$

Built upon Definition 3.1, 4.1, and Eq. 2, we are ready to state the following theorem (proof in Appendix C.1).

**Theorem 4.1.** *With the pseudo gradient $\Delta$ obtained by $\mathbf{L}(\mathcal{D}_m)$, the generalization contribution is lower bounded:*

$$\mathbb{E}_{\Delta:\mathbf{L}(\mathcal{D}_m)} W_d(\rho(\theta+\Delta), \mathcal{D}\backslash\mathcal{D}_m) \geq \mathbb{E}_{\Delta:\mathbf{L}(\mathcal{D}_m)} W_d(\rho(\theta+\Delta), \tilde{\mathcal{D}}_m) - \big| \mathbb{E}_{\Delta:\mathbf{L}(\mathcal{D}_m)} W_d(\rho(\theta+\Delta), \mathcal{D}_m)$$
$$- W_d(\rho(\theta+\Delta), \tilde{\mathcal{D}}_m) \big| - 2C_d(\mathcal{D}_m, \mathcal{D}\backslash\mathcal{D}_m),$$

*where $\tilde{\mathcal{D}}_m$ represents the dataset sampled from $\mathcal{D}_m$.*

---

[2]The similar definition is used in the literature (Franceschi et al., 2018).

[3]The pseudo gradient at round $r$ is calculated as: $\Delta^r = \theta_T^{r-1} - \theta_0^{r-1}$ with the maximum local iterations $T$.

*Remark* 4.1. Theorem 4.1 implies that three terms are related to the generalization contribution. The first and second terms are intuitive, showing that the generalization contribution of a distribution $\mathcal{D}_m$ is expected to be large and similar to that of a training dataset $\tilde{\mathcal{D}}_m$. The last term is also intuitive, which implies that promoting the generalization performance requires constructing similar conditional distributions. Both the Definition 4.1 and Theorem 4.1 use distributions conditioned on the label $y$, so we write the feature distribution $\mathcal{H}|y$ as $\mathcal{H}$ for brevity in rest of the paper.

Built upon the theoretical analysis, it is straightforward to make all client models trained on similar distributions to obtain higher generalization performance. However, collecting data to construct such a distribution is forbidden in FL due to privacy concerns. To address this challenge, we propose decoupling a deep neural network into a feature extractor network $\varphi_{\theta_{low}}$ parameterized by $\theta_{low} \in \mathbb{R}^{d_l}$ and a classifier network parameterized by $\theta_{high} \in \mathbb{R}^{d_h}$, and making the classifier network trained on the similar conditional distributions $\mathcal{H}|y$ with less discrepancy, as shown in Figure 1. Here, $d_l$ and $d_h$ represent the dimensions of parameters $\theta_{low}$ and $\theta_{high}$, respectively.

Specifically, client $m$ can estimate its own hidden feature distribution as $\mathcal{H}_m$ using the local hidden features $h = \varphi_{\theta_{low}}(x)|_{(x,y)\sim\mathcal{D}_m}$ and send $\mathcal{H}_m$ to the server for the global distribution approximation. Then, the server aggregates the received distributions to obtain the global feature distribution $\mathcal{H}$ and broadcasts it, being similar to the model average in the FedAvg. Finally, classifier networks of all clients thus performs local training on both the local hidden features $h_{(x,y)\sim\mathcal{D}_m}$ and the shared $\mathcal{H}$ during the local training. To protect privacy and reduce computation and communication costs, we propose an approximate but efficient feature estimation methods in Section 4.3. To verify the privacy protection effect, following (Luo et al., 2021), we reconstruct the raw images from features by model inversion attack (Zhao et al., 2021; Zhou et al., 2023) in Figure 11, 12 and 13 in Appendix F.7, showing FedImpro successfully protect the original data privacy.

In what follows, we show that such a decoupling strategy can reduce the gradient dissimilarity, besides the promoted generalization performance. To help understand the theory, We also provide an overview of interpreting and connecting our theory to the FedImpro in Appendix C.4.

## 4.2 DECOUPLED GRADIENT DISSIMILARITY

The gradient dissimilarity in FL resulted from heterogeneous data, i.e., the data distribution on client $m$, $\mathcal{D}_m$, is different from that on client $k$, $\mathcal{D}_k$ (Karimireddy et al., 2020). The commonly used quantitative measure of gradient dissimilarity is defined as inter-client gradient variance (CGV).

**Definition 4.2.** Inter-client Gradient Variance (CGV): (Kairouz et al., 2019; Karimireddy et al., 2020; Woodworth et al., 2020; Koloskova et al., 2020) $\text{CGV}(F, \theta) = \mathbb{E}_{(x,y)\sim\mathcal{D}_m}||\nabla f_m(\theta; x, y) - \nabla F(\theta)||^2$. CGV is usually assumed to be upper bounded (Kairouz et al., 2019; Woodworth et al., 2020; Lian et al., 2017), i.e., $\text{CGV}(F, \theta) = \mathbb{E}_{(x,y)\sim\mathcal{D}_m}||\nabla f_m(\theta; x, y) - \nabla F(\theta)||^2 \leq \sigma^2$ with a constant $\sigma$.

Upper bounded gradient dissimilarity benefits the theoretical convergence rate (Woodworth et al., 2020). Specifically, lower gradient dissimilarity directly causes higher convergence rate (Karimireddy et al., 2020; Li et al., 2020b; Woodworth et al., 2020). This means that the decoupling strategy can also benefit the convergence rate if the gradient dissimilarity can be reduced. Now, we are ready to demonstrate how to reduce the gradient dissimilarity CGV with our decoupling strategy. With representing $\nabla f_m(\theta; x, y)$ as $\{\nabla_{\theta_{low}} f_m(\theta; x, y), \nabla_{\theta_{high}} f_m(\theta; x, y)\}$, we propose that the CGV can be divided into two terms of the different parts of $\theta$ (see Appendix C.2 for details):

$$\text{CGV}(F, \theta) = \mathbb{E}_{(x,y)\sim\mathcal{D}_m}||\nabla f_m(\theta; x, y) - \nabla F(\theta)||^2 \tag{3}$$

$$= \mathbb{E}_{(x,y)\sim\mathcal{D}_m}\left[||\nabla_{\theta_{low}} f_m(\theta; x, y) - \nabla_{\theta_{low}} F(\theta)||^2 + ||\nabla_{\theta_{high}} f_m(\theta; x, y) - \nabla_{\theta_{high}} F(\theta)||^2\right].$$

According to the chain rule of the gradients of a deep model, we can derive that the high-level part of gradients that are calculated with the raw data and labels $(x, y) \sim \mathcal{D}_m$ is equal to gradients with the hidden features and labels $(h = \varphi_{\theta_{low}}(x), y)$ (proof in Appendix C.2):

$$\nabla_{\theta_{high}} f_m(\theta; x, y) = \nabla_{\theta_{high}} f_m(\theta; h, y),$$

$$\nabla_{\theta_{high}} F(\theta) = \sum_{m=1}^{M} p_m \mathbb{E}_{(x,y)\sim\mathcal{D}_m} \nabla_{\theta_{high}} f(\theta; h, y), \tag{4}$$

in which $f_m(\theta; h, y)$ is computed by forwarding the $h = \varphi_{\theta_{low}}(x)$ through the high-level model without the low-level part. In `FedImpro`, with shared $\mathcal{H}$, client $m$ will sample $\hat{h} \sim \mathcal{H}$ and

$h_m = \varphi_{\theta_{low}}(x)|_{(x,y)\sim\mathcal{D}_m}$ to train their classifier network, then the objective function becomes as [4]:

$$\min_{\theta\in\mathbb{R}^d} \hat{F}(\theta) := \sum_{m=1}^{M} \hat{p}_m \mathbb{E}_{\substack{(x,y)\sim\mathcal{D}_m \\ \hat{h}\sim\mathcal{H}}} \hat{f}(\theta; x, \hat{h}, y) \triangleq \sum_{m=1}^{M} \hat{p}_m \mathbb{E}_{\substack{(x,y)\sim\mathcal{D}_m \\ \hat{h}\sim\mathcal{H}}} \left[ f(\theta; \varphi_{\theta_{low}}(x), y) + f(\theta; \hat{h}, y) \right]. \quad (5)$$

Here, $\hat{p}_m = \frac{n_m + \hat{n}_m}{N + \hat{N}}$ with $n_m$ and $\hat{n}_m$ being the sampling size of $(x, y) \sim \mathcal{D}_m$ and $\hat{h} \sim \mathcal{H}$ respectively, and $\hat{N} = \sum_{m=1}^{M} \hat{n}_m$. Now, we are ready to state the following theorem of reducing gradient dissimilarity by sampling features from the same distribution (proof in Appendix C.3).

**Theorem 4.2.** *Under the gradient variance measure CGV (Definition 4.2), with $\hat{n}_m$ satisfying $\frac{\hat{n}_m}{n_m + \hat{n}_m} = \frac{\hat{N}}{N + \hat{N}}$, the objective function $\hat{F}(\theta)$ causes a tighter bounded gradient dissimilarity, i.e. $CGV(\hat{F}, \theta) = \mathbb{E}_{(x,y)\sim\mathcal{D}_m} ||\nabla_{\theta_{low}} f_m(\theta; x, y) - \nabla_{\theta_{low}} F(\theta)||^2 + \frac{N^2}{(N+\hat{N})^2} ||\nabla_{\theta_{high}} f_m(\theta; x, y) - \nabla_{\theta_{high}} F(\theta)||^2 \leq CGV(F, \theta)$.*

*Remark* 4.2. Theorem 4.2 shows that the high-level gradient dissimilarity can be reduced as $\frac{N^2}{(N+\hat{N})^2}$ times by sampling the same features between clients. Hence, estimating and sharing feature distributions is the key to promoting the generalization contribution and the reduction of gradient dissimilarity. Note that choosing $\hat{N} = \infty$ can eliminate high-level dissimilarity. However, two reasons make it impractical to sample infinite features $\hat{h}$. First, the distribution is estimated using limited samples, leading to biased estimations. Second, infinite sampling will dramatically increase the calculating cost. We set $\hat{N} = N$ in our experiments.

## 4.3 TRAINING PROCEDURE

As shown in Algorithm 1, `FedImpro` merely requires two extra steps compared with the vanilla FedAvg method and can be easily plugged into other FL algorithms: a) estimating and broadcasting a global distribution $\mathcal{H}$; b) performing local training with both the local data $(x, y)$ and the hidden features $(\hat{h} \sim \mathcal{H}|y, y)$.

Moreover, sampling $\hat{h} \sim \mathcal{H}|y$ has two additional advantages as follows. First, directly sharing the raw hidden features may incur privacy concerns. The raw data may be reconstructed by feature inversion methods (Zhao et al., 2021). One can use different distribution approximation methods to estimate $\{h_m | m \in \mathcal{M}\}$ to avoid exposing the raw data. Second, the hidden features usually have much higher dimensions than the raw data (Lin et al., 2021). Hence, communicating and saving them between clients and servers may not be practical. We can use different distribution approximation methods to obtain $\mathcal{H}$. Transmitting the parameters of $\mathcal{H}$ can consume less communication resource than hidden features $\{h_m | m \in \mathcal{M}\}$.

Following previous work (Kendall & Gal, 2017), we exploit the Gaussian distribution to approximate the feature distributions. Although it is an inexact estimation of real features, Gaussian distribution is computation-friendly. And the mean and variance of features used to estimate it widely exist in BatchNorm layers (Ioffe & Szegedy, 2015), which are also communicated between server and clients.

---

**Algorithm 1** Framework of FedImpro.

**server input:** initial $\theta^0$, maximum communication round $R$

**client $m$'s input:** local iterations $T$

  **Initialization:** server distributes the initial model $\theta^0$ to all clients, and the initial global $\mathcal{H}^0$ .

  **Server_Executes:**
  **for** each round $r = 0, 1, \cdots, R$ **do**
    server samples a set of clients $\mathcal{S}_r \subseteq \{1, ..., M\}$.
    server **communicates** $\theta_r$ and $\mathcal{H}^r$ to all clients $m \in \mathcal{S}$.
    **for** each client $m \in \mathcal{S}^r$ **in parallel do do**
      $\theta_{m,E-1}^{r+1}, \mathcal{H}_m^{r+1} \leftarrow \text{ClientUpdate}(m, \theta^r, \mathcal{H}^r)$.
    **end for**
    $\theta^{r+1} \leftarrow \sum_{m=1}^{M} p_m \theta_{m,E-1}^{r+1}$.
    Update $\mathcal{H}^{r+1}$ using $\{\mathcal{H}_m^{r+1} | m \in \mathcal{S}^r\}$.
  **end for**

  **ClientUpdate**$(m, \theta, \mathcal{H})$:
  **for** each local iteration $t$ with $t = 0, \cdots, T-1$ **do**
    Sample raw data $(x, y) \sim \mathcal{D}_m$ and $\hat{h} \sim \mathcal{H}|y$.
    $\theta_{m,t+1} \leftarrow \theta_{m,t} - \eta_{m,t}\nabla_\theta \hat{f}(\theta; x, \hat{h}, y)$ (Eq. 5)
    Update $\mathcal{H}_m$ using $\hat{h}_m = \varphi_{\theta_{low}}(x)$.
  **end for**
  **Return** $\theta$ and $\mathcal{H}_m$ to server.

---

[4]We reuse $f$ here for brevity, the input of $f$ can be the input $x$ or the hidden feature $h = \varphi_{\theta_{low}}(x)$.

Table 1: Best test accuracy (%) of all experimental results. "Cent." means centralized training. "Acc." means the test accuracy. Each experiment is repeated 3 times with different random seed. The standard deviation of each experiment is shown in its mean value.

| Dataset | Cent. Acc. | FL Setting | | | FL Test Accuracy | | | | | |
|---------|------------|---|---|---|---------|---------|----------|---------|--------|----------|
| | | $a$ | $E$ | $M$ | FedAvg | FedProx | SCAFFOLD | FedNova | FedDyn | **FedImpro** |
| CIFAR-10 | 92.53 | 0.1 | 1 | 10 | 83.65±2.03 | 83.22±1.52 | 82.33±1.13 | 84.97±0.87 | 84.73±0.81 | **88.45±0.43** |
| | | 0.05 | 1 | 10 | 75.36±1.92 | 77.49±1.24 | 33.6±3.87 | 73.49±1.42 | 77.21±1.25 | **81.75±1.03** |
| | | 0.1 | 5 | 10 | 85.69±0.57 | 85.33±0.39 | 84.4±0.41 | 86.92±0.28 | 86.59±0.39 | **88.10±0.20** |
| | | 0.1 | 1 | 100 | 73.42±1.19 | 68.59±1.03 | 59.22±3.11 | 74.94±0.98 | 75.29±0.95 | **77.56±1.02** |
| | | Average | | | 79.53 | 78.66 | 64.89 | 80.08 | 80.96 | **83.97** |
| FMNIST | 93.7 | 0.1 | 1 | 10 | 88.67±0.34 | 88.92±0.25 | 87.81±0.36 | 87.97±0.41 | 89.01±0.25 | **90.83±0.19** |
| | | 0.05 | 1 | 10 | 82.73±0.98 | 83.66±0.82 | 76.16±1.29 | 81.89±0.91 | 83.20±1.19 | **86.42±0.71** |
| | | 0.1 | 5 | 10 | 87.6±0.52 | 88.41±0.38 | 88.44±0.29 | 87.66±0.62 | 88.50±0.52 | **89.87±0.21** |
| | | 0.1 | 1 | 100 | 90.12±0.19 | 90.39±0.12 | 88.24±0.31 | 90.40±0.18 | 90.57±0.21 | **90.98±0.15** |
| | | Average | | | 87.28 | 87.85 | 85.16 | 86.98 | 87.82 | **89.53** |
| SVHN | 95.27 | 0.1 | 1 | 10 | 88.20±1.21 | 87.04±0.89 | 83.87±2.15 | 88.48±1.31 | 90.82±1.09 | **92.37±0.82** |
| | | 0.05 | 1 | 10 | 80.67±1.92 | 82.39±1.35 | 82.29±1.81 | 84.01±1.48 | 84.12±1.28 | **90.25±0.69** |
| | | 0.1 | 5 | 10 | 86.32±1.19 | 86.05±0.72 | 83.14±1.51 | 88.10±0.91 | 89.92±0.50 | **91.58±0.72** |
| | | 0.1 | 1 | 100 | 92.42±0.21 | 92.29±0.18 | 92.06±0.20 | 92.44±0.31 | 92.82±0.13 | **93.42±0.15** |
| | | Average | | | 86.90 | 86.94 | 85.34 | 88.26 | 89.42 | **91.91** |
| CIFAR-100 | 74.25 | 0.1 | 1 | 10 | 69.38±1.02 | 69.78±0.91 | 65.74±1.52 | 69.52±0.75 | 69.59±0.52 | **70.28±0.33** |
| | | 0.05 | 1 | 10 | 63.80±1.29 | 64.75±1.25 | 61.49±2.16 | 64.57±1.27 | 64.90±0.69 | **66.60±0.91** |
| | | 0.1 | 5 | 10 | 68.39±1.02 | 68.71±0.88 | 68.67±1.20 | 67.99±1.04 | 68.52±0.41 | **68.79±0.52** |
| | | 0.1 | 1 | 100 | 53.22±1.20 | 54.10±1.32 | 23.77±3.21 | 55.40±0.81 | 54.82±0.81 | **56.07±0.75** |
| | | Average | | | 63.70 | 64.34 | 54.92 | 64.37 | 64.46 | **65.44** |

On each client $m$, a Gaussian distribution $\mathcal{N}(\mu_m, \sigma_m)$ parameterized with $\mu_m$ and $\sigma_m$ is used to approximate the feature distribution. On the server-side, $\mathcal{N}(\mu_g, \sigma_g)$ estimate the global feature distributions. As shown in Figure 1 and Algorithm 1, during the local training, clients update $\mu_m$ and $\sigma_m$ using the real feature $h_m$ following a moving average strategy which is widely used in the literature (Ioffe & Szegedy, 2015; Wang et al., 2021):

$$\mu_m^{(t+1)} = \beta_m \mu_m^{(t)} + (1 - \beta_m) \times \text{mean}(h_m),$$
$$\sigma_m^{(t+1)} = \beta_m \sigma_m^{(t)} + (1 - \beta_m) \times \text{variance}(h_m), \tag{6}$$

where $t$ is the iteration of the local training, $\beta_m$ is the momentum coefficient. To enhance privacy protection, on the server side, $\mu_g$ and $\sigma_g$ are updated with local parameters plus noise $\epsilon_i^r$:

$$\mu_g^{(r+1)} = \beta_g \mu_g^{(r)} + (1 - \beta_g) \times \frac{1}{|\mathcal{S}^r|} \sum_{i \in \mathcal{S}^r} (\mu_i^r + \epsilon_r^r),$$
$$\sigma_g^{(r+1)} = \beta_g \sigma_g^{(r)} + (1 - \beta_g) \times \frac{1}{|\mathcal{S}^r|} \sum_{i \in \mathcal{S}^r} (\sigma_i^r + \epsilon_i^r), \tag{7}$$

where $\epsilon_i^r \sim \mathcal{N}(0, \sigma_\epsilon)$. Appendix F.2 and Table F.2 show the performance of FedImpro with different noise degrees $\sigma_\epsilon$. And Appendix F.7 show that the model inversion attacks fail to invert raw data based on shared $\mu_m, \sigma_m, \mu_g$ and $\sigma_g$, illustrating that FedImpro can successfully protect the original data privacy.

## 5 EXPERIMENTS

### 5.1 EXPERIMENT SETUP

**Federated Datasets and Models.** We verify FedImpro with four datasets commonly used in the FL community, i.e., CIFAR-10 (Krizhevsky & Hinton, 2009), FMNIST (Xiao et al., 2017), SVHN (Netzer et al., 2011), and CIFAR-100 (Krizhevsky & Hinton, 2009). We use the Latent Dirichlet Sampling (LDA) partition method to simulate the Non-IID data distribution, which is the most used partition method in FL (He et al., 2020b; Li et al., 2021c; Luo et al., 2021). We train Resnet-18 on CIFAR-10, FMNIST and SVHN, and Resnet-50 on CIFAR-100. We conduct experiments with two different Non-IID degrees, $a = 0.1$ and $a = 0.05$. We simulate cross-silo FL with $M = 10$ and cross-device FL with $M = 100$. To simulate the partial participation in each round, the number of sample clients is 5 for $M = 10$ and 10 for $M = 100$. Some additional experiment results are shown in Appendix F

**Baselines and Metrics.** We choose the classical FL algorithm, FedAvg (McMahan et al., 2017), and recent effective FL algorithms proposed to address the client drift problem, including FedProx (Li et al., 2020b), SCAFFOLD (Karimireddy et al., 2020), and FedNova (Wang et al., 2020a), as our baselines. The detailed hyper-parameters of all experiments are reported in Appendix E. We use two metrics, the best accuracy and the number of communication rounds to achieve a target accuracy, which is set to the best accuracy of FedAvg. We also measure the weight divergence (Karimireddy et al., 2020), $\frac{1}{|S^r|} \sum_{i \in S^r} \|\bar{\theta} - \theta_i\|$, as it reflects the effect on gradient dissimilarity reduction[5].

## 5.2 EXPERIMENTAL RESULTS

**Basic FL setting.** As shown in Table 1, using the classical FL training setting, i.e. $a = 0.1$, $E = 5$ and $M = 10$, for CIFAR-10, FMNIST and SVHN, FedImpro achieves much higher generalization performance than other methods. We also find that, for CIFAR-100, the performance of FedImpro is similar to FedProx. We conjecture that CIFAR-100 dataset has more classes than other datasets, leading to the results. Thus, a powerful feature estimation approach instead of a simple Gaussian assumption can be a promising direction to enhance the performance.

**Impacts of Non-IID Degree.** As shown in Table 1, for all datasets with high Non-IID degree ($a = 0.05$), FedImpro obtains more performance gains than the case of lower Non-IID degree ($a = 0.1$). For example, we obtain 92.37% test accuracy on SVHN with $a = 0.1$, higher than the FedNova by 3.89%. Furthermore, when Non-IID degree increases to $a = 0.05$, we obtain 90.25% test accuracy, higher than FedNova by 6.14%. And for CIFAR-100, FedImpro shows benefits when $a = 0.05$, demonstrating that FedImpro can defend against more severe data heterogeneity.

**Different Number of Clients.** We also show the results of 100-client FL setting in Table 1. FedImpro works well with all datasets, demonstrating excellent scalability with more clients.

**Different Local Epochs.** More local training epochs $E$ could reduce the communication rounds, saving communication cost in practical scenarios. In Table 1, the results of $E = 5$ on CIFAR-10, FMNIST, and SVHN verify that FedImpro works well when increasing local training time.

**Weight Divergence.** We show the weight divergence of different methods in Figure 2 (b), where FedNova is excluded due to its significant weight divergence. The divergence is calculated by $\frac{1}{|S^r|} \sum_{i \in S^r} \|\bar{\theta} - \theta_i\|$, which shows the dissimilarity of local client models after local training. At the initial training stage, the weight divergence is similar for different methods. During this stage, the low-level model is still unstable and the feature estimation is not accurate. After about 500 communication rounds, FedImpro shows lower weight divergence than others, indicating that it converges faster than others.

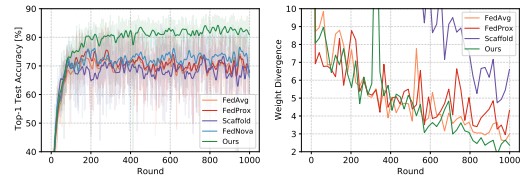

(a) Test Accuracy     (b) Weight Divergence
Figure 2: CIFAR10 with $a = 0.1$, $E = 1$, $M = 10$.

**Convergence Speed.** Figure 2 (a) shows that FedImpro can accelerate the convergence of FL.[6] And we compare the communication rounds that different algorithms need to attain the target accuracy in Table 2. Results show FedImpro significantly improves the convergence speed.

Table 3: Splitting at different convolution layers in ResNet-18.

| Layer | 5-th | 9-th | 13-th | 17-th |
|---|---|---|---|---|
| Test Acc. (%) | 87.64 | 88.45 | 87.86 | 84.08 |

## 5.3 ABLATION STUDY

To verify the impacts of the depth of gradient decoupling, we conduct experiments by splitting at different layers, including the 5-th, 9-th, 13-th and 17-th layers. Table 3 demonstrates that FedImpro can obtain benefits at low or middle layers. Decoupling at the 17-th layer will decrease the performance, which is consistent with our conclusion in Sec. 4.2. Specifically, decoupling at a

---

[5]To ensure the reproducibility of experimental results, we open source our code at `https://github.com/wizard1203/FedImpro`.

[6]Due to high instability of training with severe heterogeneity, the actual test accuracy are semitransparent lines and the smoothed accuracy are opaque lines for better visualization. More results in Appendix E.3.

Table 2: Communication Round to attain the target accuracy. "NaN." means not achieving the target.

| Dataset | FL Setting | | | Target Acc. | Communication Round to attain the target accuracy | | | | | |
|---|---|---|---|---|---|---|---|---|---|---|
| | $a$ | $E$ | $M$ | | FedAvg | FedProx | SCAFFOLD | FedNova | FedDyn | **FedImpro** |
| CIFAR-10 | 0.1 | 1 | 10 | 82.0 | 142 | **128** | 863 | 142 | 291 | **128** |
| | 0.05 | 1 | 10 | 73.0 | 247 | 121 | NaN | 407 | 195 | **112** |
| | 0.1 | 5 | 10 | 84.0 | 128 | 128 | 360 | 80 | 109 | **78** |
| | 0.1 | 1 | 100 | 73.0 | 957 | NaN | NaN | 992 | 820 | **706** |
| FMNIST | 0.1 | 1 | 10 | 87.0 | 83 | 76 | 275 | 83 | 62 | **32** |
| | 0.05 | 1 | 10 | 81.0 | 94 | 94 | NaN | 395 | 82 | **52** |
| | 0.1 | 5 | 10 | 87.0 | 147 | 31 | 163 | 88 | 29 | **17** |
| | 0.1 | 1 | 100 | 90.0 | 375 | 470 | NaN | **317** | 592 | 441 |
| SVHN | 0.1 | 1 | 10 | 87.0 | 292 | 247 | NaN | 251 | 162 | **50** |
| | 0.05 | 1 | 10 | 80.0 | 578 | 68 | 358 | 242 | 120 | **50** |
| | 0.1 | 5 | 10 | 86.0 | 251 | 350 | NaN | NaN | 92 | **11** |
| | 0.1 | 1 | 100 | 92.0 | 471 | 356 | 669 | 356 | 429 | **346** |
| CIFAR-100 | 0.1 | 1 | 10 | 69.0 | 712 | 857 | NaN | 733 | 682 | **614** |
| | 0.05 | 1 | 10 | 61.0 | 386 | 386 | 755 | 366 | 416 | **313** |
| | 0.1 | 5 | 10 | 68.0 | 335 | 307 | **182** | 282 | 329 | 300 |
| | 0.1 | 1 | 100 | 53.0 | 992 | 939 | NaN | 910 | 951 | **854** |

very high layer may not be enough to resist gradient dissimilarity, leading to weak data heterogeneity mitigation. Interestingly, according to Theorem 4.2, decoupling at the 5-th layer should diminish more gradient dissimilarity than the 9-th and 13-th layers; but it does not show performance gains. We conjecture that it is due to the difficulty of distribution estimation, since biased estimation leads to poor generalization contribution. As other works (Lin et al., 2021) indicate, features at the lower level usually are richer larger than at the higher level. Thus, estimating the lower-level features is much more difficult than the higher-level.

## 5.4 DISCUSSION

In this section, we provide some more experimental supports for FedImpro. All experiment results of this section are conducted on CIFAR-10 with ResNet-18, $a = 0.1$, $E = 1$ and $M = 10$. And further experiment result are shown in Appendix F due to the limited space.

FedImpro only guarantees the reduction of high-level gradient dissimilarity without considering the low-level part. We experimentally find that low-level weight divergence shrinks faster than

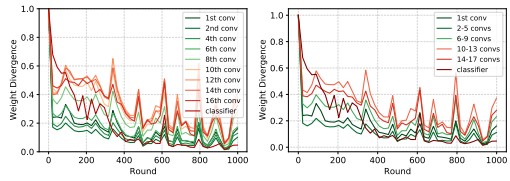

(a) Some Choosen layers.  (b) Different Parts.

Figure 3: Layer divergence of FedAvg.

high-level. Here, we show the layer-wise weight divergence in Figure 3. We choose and show the divergence of 10 layers in Figure 3 (a), and the different stages of ResNet-18 in Figure 3 (b). As we hope to demonstrate the divergence trend, we normalize each line with its maximum value. The results show that the low-level divergence shrinks faster than the high-level divergence. This means that reducing the high-level gradient dissimilarity is more important than the low-level.

## 6 CONCLUSION

In this paper, we correct client drift from a novel perspective of generalization contribution, which is bounded by the conditional Wasserstein distance between clients' distributions. The theoretical conclusion inspires us to propose decoupling neural networks and constructing similar feature distributions, which greatly reduces the gradient dissimilarity by training with a shared feature distribution without privacy breach. We theoretically verify the gradient dissimilarity reduction and empirically validate the benefits of `FedImpro` on generalization performance. Our work opens a new path of enhancing FL performance from a generalization perspective. Future works may exploit better feature estimators like generative models (Goodfellow et al., 2014; Karras et al., 2019) to sample higher-quality features, while reducing the communication and computation costs.

## 7 ACKNOWLEDGMENT

This work was partially supported by National Natural Science Foundation of China under Grant No. 62272122, a Hong Kong RIF grant under Grant No. R6021-20, and Hong Kong CRF grants under Grant No. C2004-21G and C7004-22G. TL is partially supported by the following Australian Research Council projects: FT220100318, DP220102121, LP220100527, LP220200949, IC190100031. BH was supported by the NSFC General Program No. 62376235, Guangdong Basic and Applied Basic Research Foundation No. 2022A1515011652, HKBU Faculty Niche Research Areas No. RC-FNRA-IG/22-23/SCI/04, and CCF-Baidu Open Fund. XMT was supported in part by NSFC No. 62222117, the Fundamental Research Funds for the Central Universities under contract WK3490000005, and KY2100000117. SHS was supported in part by the National Natural Science Foundation of China (NSFC) under Grant No. 62302123 and Guangdong Provincial Key Laboratory of Novel Security Intelligence Technologies under Grant 2022B1212010005.

We thank the area chair and reviewers for their valuable comments.

## 8 ETHICS STATEMENT

This paper does not raise any ethics concerns. This study does not involve any human subjects, practices to data set releases, potentially harmful insights, methodologies and applications, potential conflicts of interest and sponsorship, discrimination/bias/fairness concerns, privacy and security issues, legal compliance, and research integrity issues.

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

SUPPLEMENTARY MATERIAL

## A   BROADER IMPACT

**Measuring Client Contribution During Local Training.** As discussed in the section 2, current works mainly focus on measuring generalization contribution from clients from participating during the whole training process. We consider measuring this contribution during each communication round, which opens a new angle toward the convergence analysis of FL. Future works may fill the generalization gap between FL and centralized training with all datasets.

**Relationship between Privacy and Performance.** We analyze the relationship between the sharing features and the raw data in section 4.2 and Appendix. However, we do not deeply investigate how sharing features or parameters of estimated feature distribution threatens the privacy of private raw data. Sharing features at a lower level may reduce gradient dissimilarity and high generalization performance of FL, yet leading to higher risks of data privacy. Future works may consider figuring out the trade-off between data privacy and the generalization performance with sharing features.

**Connections of our work to knowledge distillation and domain generalization.** The approximation of features generated based on the client data and low-level models can be seen as a kind of knowledge distillation of other clients. More in-depth analyses of this problem would be an exciting direction, which will be added to our future works. The domain generalization is also an exciting connection to federated learning. It is interesting to connect the measurements of client contribution to the domain generalization.

## B   MORE DISCUSSIONS

**Extra computational overhead.** The experiment hardware and software are described in Section E.1. FedImpro requires little more computing time of FedAvg in simulation. Almost all current experiments are usually simulated in FL community  (He et al., 2020b; Lai et al., 2022; Li et al., 2021d). We provide a quantitative comparison between FedAvg and FedImpro based on this test setup: Training ResNet-18 on Dirichlet partitioned CIFAR-10 dataset with $\alpha = 0.1$, $M = 10$, sampling 5 clients each round, with total communication rounds as 1000 and local epoch as 1.

The simulation time of the FedAvg is around 8h 52m 12s, while FedImpro consumes around 10h 32m 32s. According to the Table 2 in the main text, FedImpro achieves the 82% acc in similar cpu time with FedAvg. When $E = 5$, the simulation time of FedAvg increases to 20h 10m 25s. The convergence speed of FedImpro with $E = 1$ is better than FedAvg with $E = 5$. Note that in this simulation, the communication time is ignored, because the real-world communication does not happen.

**Extra communication overhead.** Communiation round is a more important real-world metric compared to cpu time in simulation. In real-world federated learning, the bandwidth with the Internet (around $1 \sim 10$ MB/s) is very low comparing to the cluster environemnt (around $10 \sim 1000$ GB/s). Taking ResNet18 with 48 MB as example, each communication with 10 clients would consume $48 \sim 480$s, 1000 rounds requires $48000 \sim 480000$s ($15 \sim 150$ hours). And some other factors like the communication latency and unstability will further increase the communication time. Thus, the large costs mainly exist in communication overhead instead of the computing time (McMahan et al., 2017; Kairouz et al., 2019). As shown in Table 2 in original text, FedImpro achieve mush faster convergence than other algorithms.

The communication time in real-world can be characterized by the alpha-beta model (Thakur et al., 2005; Shi et al., 2019): $T_{comm} = \alpha + \beta M$, where $\alpha$ is the latency per message (or say each communication), $\beta$ the inverse of the communication speed, and $M$ the message size. In FedImpro, the estimated parameters are communicated along with the model parameters. Thus (a) the number of communications is not increased, and the communication time from $\alpha$ will not increase; (b) The $M$ increased as $M_{model} + M_{estimator}$, where $M_{model}$ is the model size and $M_{estimator}$ is the size of estimated parameters. Taking ResNet-20 as an example, the size of estimated parameters is equal to double (mean and variance) feature size (not increased with batch size) as 4KB (ResNet-20), which is greatly less than the model size around 48 MB. Therefore, the additional communication cost is almost zero.

**Does the low-level model benefits from the $\hat{h}$?** The low-level model will not receive gradients calculated from the $\hat{h}$. Thus, the low-level model is not explicitly updated by the $\hat{h}$. However, from another aspect, training the high-level model benefits from the closer-distance features and the reduced gradient heterogeneity. In FedImpro, the backward-propagated gradients on original features $h$ have lower bias than FedAvg. Therefore, low-level model can implicitly benefit from $\hat{h}$.

It is non-trivial to conduct the ablation study on training low-level model with $\hat{h}$. The implicit benefits, i.e. better backward gradients from the high-level models, require the high-level models are also better ("better" means that they are good for model generalization and defending data heterogeneity). Thus, the benefits on high-level and low-level model are coupled. Thus, how to build a direct bridge to benefit training low-level model from $\hat{h}$ is still under exploring.

**When label distribution is the same on different clients, can we still benefit from FedImpro?** In this case, the client drift may still happen due to the feature distribution is different (Li et al., 2021b; Kairouz et al., 2019). In this case, we may utilize the Wasserstein distance not conditioned on label in unsupervised domain adaptation (Lee et al., 2019; Li et al., 2020a) to analyse this problem. We will consider this case as the future work.

**FedAvging on the low-level model or keeping the low-level model locally for each clients?** FedImpro focuses on how to better learning a global model instead of personalized or heterogeneous models. Thus, we conduct FedAvg on the low-level model. However, The core idea of FedImpro, sharing estimated features would be very helpful in personalized split-FL (keep the low-level model locally), with these reasons: (a) our Theorem 4.1 and 4.2 can also be applied into personalized FL with sharing estimated features; (b) The large communication overhead in personalized split-FL is large due to the communication in each forward and backward propagation (low-level and high-level models are typically deployed in clients and server, respectively (Liang et al., 2020; Collins et al., 2021; Thapa et al., 2020; Chen & Chao, 2021)). Now, sharing estimated features would help reduce this communication a lot by decoupling the forward and backward propagations.

## C  PROOF

### C.1  BOUNDED GENERALIZATION CONTRIBUTION

Given client $m$, we quantify the generalization contribution, in FL systems as follows:

$$\mathbb{E}_{\Delta:\mathbf{L}(\mathcal{D}_m)}W_d(\rho(\theta + \Delta), \mathcal{D}\backslash\mathcal{D}_m)), \tag{8}$$

where $\Delta$ is a pseudo gradient obtained by applying a learning algorithm $\mathbf{L}(\mathcal{D}_m)$ to a distribution $\mathcal{D}_m$, $W_d$ is the quantification of generalization, and $\mathcal{D}\backslash\mathcal{D}_m)$ means the distribution of all clients except for client $m$.

**Theorem C.1.** *With the pseudo gradient $\Delta$ obtained by $\mathbf{L}(\mathcal{D}_m)$, the generalization contribution is lower bounded:*

$$\mathbb{E}_{\Delta:\mathbf{L}(\mathcal{D}_m)}W_d(\rho(\theta + \Delta), \mathcal{D}\backslash\mathcal{D}_m)) \geq \mathbb{E}_{\Delta:\mathbf{L}(\mathcal{D}_m)}W_d(\rho(\theta + \Delta), \tilde{\mathcal{D}}_m) - |\mathbb{E}_{\Delta:\mathbf{L}(\mathcal{D}_m)}W_d(\rho(\theta + \Delta), \mathcal{D}_m))$$
$$- W_d(\rho(\theta + \Delta), \tilde{\mathcal{D}}_m))| - 2C_d(\mathcal{D}_m, \mathcal{D}\backslash\mathcal{D}_m),$$

*where $\tilde{\mathcal{D}}_m$ represents the dataset sampled from $\mathcal{D}_m$.*

*Proof.* To derive the lower bound, we decompose the conditional quantification of generalization, i.e., $W_d(\rho(\theta + \Delta), \mathcal{D}\backslash\mathcal{D}_m)$:

$$W_d(\rho(\theta + \Delta), \mathcal{D}\backslash\mathcal{D}_m) = W_d(\rho(\theta + \Delta), \mathcal{D}\backslash\mathcal{D}_m) - W_d(\rho(\theta + \Delta), \mathcal{D}_m) + W_d(\rho(\theta + \Delta), \mathcal{D}_m)$$
$$- W_d(\rho(\theta + \Delta), \tilde{\mathcal{D}}_m) + W_d(\rho(\theta + \Delta), \tilde{\mathcal{D}}_m), \tag{9}$$

where we denote $\rho$ as $\rho(\theta + \Delta)$ for brevity and $\tilde{\mathcal{D}}_m$ stands for the dataset sampled from $\mathcal{D}_m$. Built upon the decomposition, we have:

$$\mathbb{E}_{\Delta:\mathbf{L}(\mathcal{D}_m)}W_d(\rho(\theta + \Delta), \mathcal{D}\backslash\mathcal{D}_m)) \geq \mathbb{E}_{\Delta:\mathbf{L}(\mathcal{D}_m)}W_d(\rho(\theta + \Delta), \tilde{\mathcal{D}}_m)$$
$$- |\mathbb{E}_{\Delta:\mathbf{L}(\mathcal{D}_m)}W_d(\rho(\theta + \Delta), \mathcal{D}_m)) - W_d(\rho(\theta + \Delta), \tilde{\mathcal{D}}_m))|$$
$$- |\mathbb{E}_{\Delta:\mathbf{L}(\mathcal{D}_m)}W_d(\rho(\theta + \Delta), \mathcal{D}\backslash\mathcal{D}_m)) - W_d(\rho(\theta + \Delta), \mathcal{D}_m))|. \tag{10}$$

The first term in Eq. 10 represents the empirical generalization performance. The second term in Eq. 10 means that the performance gap between the model trained on sampled dataset and that trained on the distribution, rigorous analysis can be found in (Montasser et al., 2019). Note that, the first two terms are independent on the distribution $\mathcal{D}\backslash\mathcal{D}_m$, so the focus of generalization contribution is mainly on the last term, i.e., $|\mathbb{E}_{\Delta:\mathbf{L}(\mathcal{D}_m)}W_d(\rho(\theta+\Delta),\mathcal{D}\backslash\mathcal{D}_m))-W_d(\rho(\theta+\Delta),\mathcal{D}_m))|$.

The proof is relatively straightforward, as long as we derive the upper bound of $W_d(\rho(\theta+\Delta),\mathcal{D}_m)$ and $W_d(\rho(\theta+\Delta),\mathcal{D}\backslash\mathcal{D}_m)$. For $W_d(\rho(\theta+\Delta),\mathcal{D}_m)$, we have:

$$
\begin{aligned}
&W_d(\rho(\theta+\Delta),\mathcal{D}_m)\\
=&\mathbb{E}_{(\cdot|y)\sim\mathcal{D}_m}\mathbb{E}_{x\sim\mathcal{D}_m|y}\inf_{\mathrm{argmax}_i\rho(\theta;x')_i\neq y}d(x,x')\\
=&\mathbb{E}_{(\cdot|y)\sim\mathcal{D}_m}\mathbb{E}_{(x,x'')\sim J_y}\inf_{\mathrm{argmax}_i\rho(\theta;x')_i\neq y}d(x,x')\\
\leq&\mathbb{E}_{(\cdot|y)\sim\mathcal{D}_m}\mathbb{E}_{(x,x'')\sim J_y}\inf_{\mathrm{argmax}_i\rho(\theta;x')_i\neq y}d(x',x'')+d(x,x'')\\
=&\mathbb{E}_{(\cdot|y)\sim\mathcal{D}_m}\mathbb{E}_{(x,x'')\sim J_y}\inf_{\mathrm{argmax}_i\rho(\theta;x')_i\neq y}d(x',x'')+\mathbb{E}_{(\cdot|y)\sim\mathcal{D}_m}\mathbb{E}_{(x,x'')\sim J_y}d(x,x'')\\
=&\mathbb{E}_{(\cdot|y)\sim\mathcal{D}_m}\mathbb{E}_{x''\sim\mathcal{D}\backslash\mathcal{D}_m|y}\inf_{\mathrm{argmax}_i\rho(\theta;x')_i\neq y}d(x',x'')+\mathbb{E}_{(\cdot|y)\sim\mathcal{D}_m}\mathbb{E}_{(x,x'')\sim J_y}d(x,x''),
\end{aligned}
$$

where $J_y$ stands for the optimal transport between the conditional distribution $\mathcal{D}_m|y$ and $\mathcal{D}\backslash\mathcal{D}_m|y$. Similarly, we have:

$$
\begin{aligned}
W_d(\rho(\theta+\Delta),\mathcal{D}\backslash\mathcal{D}_m)\leq&\mathbb{E}_{(\cdot|y)\sim\mathcal{D}\backslash\mathcal{D}_m}\mathbb{E}_{x''\sim\mathcal{D}_m|y}\inf_{\mathrm{argmax}_i\rho(\theta;x')_i\neq y}d(x',x'')\\
&+\mathbb{E}_{(\cdot|y)\sim\mathcal{D}\backslash\mathcal{D}_m}\mathbb{E}_{(x,x'')\sim J_y}d(x,x'').
\end{aligned}
$$

Combining these two inequality, we have:

$$
\begin{aligned}
|W_d(\rho(\theta+\Delta),\mathcal{D}_m)-W_d(\rho(\theta+\Delta),\mathcal{D}\backslash\mathcal{D}_m)|\leq&2C_d(\mathcal{D}_m,\mathcal{D}\backslash\mathcal{D}_m))\\
&+\max\left\{\delta(\mathcal{D}_m,\mathcal{D}\backslash\mathcal{D}_m),\gamma(\mathcal{D}_m,\mathcal{D}\backslash\mathcal{D}_m)\right\},
\end{aligned}
\tag{11}
$$

where

$$
\begin{aligned}
\delta(\mathcal{D}_m,\mathcal{D}\backslash\mathcal{D}_m)=&\mathbb{E}_{(\cdot|y)\sim\mathcal{D}_m}\mathbb{E}_{x''\sim\mathcal{D}\backslash\mathcal{D}_m|y}\inf_{\mathrm{argmax}_i\rho(\theta;x')_i\neq y}d(x',x'')\\
&-\mathbb{E}_{(\cdot|y)\sim\mathcal{D}\backslash\mathcal{D}_m}\mathbb{E}_{x''\sim\mathcal{D}\backslash\mathcal{D}_m|y}\inf_{\mathrm{argmax}_i\rho(\theta;x')_i\neq y}d(x',x''),
\end{aligned}
$$

and

$$
\begin{aligned}
\gamma(\mathcal{D}_m,\mathcal{D}\backslash\mathcal{D}_m)=&\mathbb{E}_{(\cdot|y)\sim\mathcal{D}\backslash\mathcal{D}_m}\mathbb{E}_{x''\sim\mathcal{D}_m|y}\inf_{\mathrm{argmax}_i\rho(\theta;x')_i\neq y}d(x',x'')\\
&-\mathbb{E}_{(\cdot|y)\sim\mathcal{D}_m}\mathbb{E}_{x''\sim\mathcal{D}_m|y}\inf_{\mathrm{argmax}_i\rho(\theta;x')_i\neq y}d(x',x'').
\end{aligned}
$$

The upper bound is straightforward. For example, if the label distributions are the same, i.e. $y\sim\mathcal{D}\backslash\mathcal{D}_m$ is equal to $y\sim\mathcal{D}_m$, we have:

$$
|W_d(\rho(\theta+\Delta),\mathcal{D}_m)-W_d(\rho(\theta+\Delta),\mathcal{D}\backslash\mathcal{D}_m)|\leq 2C_d(\mathcal{D}_m,\mathcal{D}\backslash\mathcal{D}_m)).
$$

According to Eq. 11, the last term in Eq. 10 is bounded:

$$
\begin{aligned}
&|\mathbb{E}_{\Delta:\mathbf{L}(\mathcal{D}_m)}W_d(\rho(\theta+\Delta),\mathcal{D}_m))-W_d(\rho(\theta+\Delta),\tilde{\mathcal{D}}_m))|\\
\leq&\mathbb{E}_{\Delta:\mathbf{L}(\mathcal{D}_m)}|W_d(\rho(\theta+\Delta),\mathcal{D}_m))-W_d(\rho(\theta+\Delta),\tilde{\mathcal{D}}_m))|,
\end{aligned}
\tag{12}
$$

which is further upper bounded by conditional Wasserstein distance when the label distributions are not the same:

$$
\begin{aligned}
&\mathbb{E}_{\Delta:\mathbf{L}(\mathcal{D}_m)}|W_d(\rho(\theta+\Delta),\mathcal{D}_m))-W_d(\rho(\theta+\Delta),\tilde{\mathcal{D}}_m))|\\
&\leq 2C_d(\mathcal{D}_m,\mathcal{D}\backslash\mathcal{D}_m))+\max\left\{\delta(\mathcal{D}_m,\mathcal{D}\backslash\mathcal{D}_m),\gamma(\mathcal{D}_m,\mathcal{D}\backslash\mathcal{D}_m)\right\}.
\end{aligned}
\tag{13}
$$

Thus, the label distribution will have additional impact on the bound. If the label distributions are the same, then we have

$$
|\mathbb{E}_{\Delta:\mathbf{L}(\mathcal{D}_m)}W_d(\rho(\theta+\Delta),\mathcal{D}_m))-W_d(\rho(\theta+\Delta),\tilde{\mathcal{D}}_m))|\leq 2C_d(\mathcal{D}_m,\mathcal{D}\backslash\mathcal{D}_m)),
\tag{14}
$$

which completes the proof. $\qquad\square$

## C.2 Derivation of Decoupling Gradient Variance

**The derivation of Equation 3.** Because $\nabla f_m = \left\{\nabla_{\theta_{low}} f_m, \nabla_{\theta_{high}} f_m\right\} \in \mathbb{R}^d$, $\nabla_{\theta_{low}} f_m \in \mathbb{R}^{d_l}$ and $\nabla_{\theta_{high}} f_m \in \mathbb{R}^{d_h}$, we have

$$\mathbb{E}_{(x,y)\sim\mathcal{D}_m}||\nabla f_m(\theta; x, y) - \nabla F(\theta)||^2 \tag{15}$$

$$=\sum_{i=1}^{d}(\nabla f_m(\theta; x, y)_{(i)} - \nabla F(\theta)_{(i)})^2$$

$$=\sum_{i=1}^{d_l}(\nabla f_m(\theta; x, y)_{(i)} - \nabla F(\theta)_{(i)})^2 + \sum_{i=d_l+1}^{d_h}(\nabla f_m(\theta; x, y)_{(i)} - \nabla F(\theta)_{(i)})^2$$

$$=\mathbb{E}_{(x,y)\sim\mathcal{D}_m}\left[||\nabla_{\theta_{low}} f_m(\theta; x, y) - \nabla_{\theta_{low}} F(\theta)||^2 + ||\nabla_{\theta_{high}} f_m(\theta; x, y) - \nabla_{\theta_{high}} F(\theta)||^2\right]$$

**The derivation of Equation 4.** Assuming a multi-layers neural network consists $L$ linear layers, each of which is followed by an activation function. And the loss function is $CE(\cdot)$. The forward function can be formulated as:

$$f(\theta, x) = CE(\tau_n(\theta_n(\tau_{n-1}(\theta_{n-1}\tau_{n-2}(...\tau_1(\theta_1 x)))))) \tag{16}$$

Then the gradient on $l$-th weight should be:

$$g_l = \frac{\partial f}{\partial \theta_l} = \frac{\partial f}{\partial \tau_n(z_n)}\frac{\partial \tau_n(z_n)}{\partial z_n}\frac{\partial z_n}{\partial \tau_{n-1}(z_{n-1})}\frac{\partial \tau_{n-1}(z_{n-1})}{\partial z_{n-1}}\frac{\partial z_{n-1}}{\partial \tau_{n-2}(z_{n-2})}...\frac{\partial \tau_{l+1}(z_{l+1})}{\partial z_{l+1}}\frac{\partial z_l}{\partial \theta_l} \tag{17}$$

$$= \frac{\partial f}{\partial \tau_n(z_n)}\tau_n'(z_n)\theta_n\tau_n'(z_{n-1})\theta_{n-1}...\tau_{l+1}'(z_{l+1})\tau_l(z_l) \tag{18}$$

$$= \frac{\partial f}{\partial \tau_n(z_n)}\left(\prod_{i=l+2}^{n}\tau_i'(z_i)\theta_i\right)\tau_{l+1}'(z_{l+1})\tau_l(z_l), \tag{19}$$

in which $\theta_l$, $\tau_l$, $z_l$, is the weight, activation function, output of the $l$-th layer, respectively. Thus, we can see that the gradient of $l$-th layer is independent of the data, hidden features, and weights before $l$-th layer if we directly input a $z_l$ to $l$-th layer.

## C.3 Proof of Theorem 4.2

We restate the optimization goals of using the private raw data $(x, y)$ of clients and the shared hidden features $\hat{h} \sim \mathcal{H}|y$ as following:

$$\min_{\theta\in\mathbb{R}^d}\hat{F}(\theta) := \sum_{m=1}^{M}\hat{p}_m\mathbb{E}_{\substack{(x,y)\sim\mathcal{D}_m\\\hat{h}\sim\mathcal{H}}}\hat{f}(\theta; x, \hat{h}, y) = \sum_{m=1}^{M}\hat{p}_m\mathbb{E}_{\substack{(x,y)\sim\mathcal{D}_m\\\hat{h}\sim\mathcal{H}}}\left[f(\theta; x, y) + f(\theta; \hat{h}, y)\right], \tag{20}$$

**Theorem C.2.** *Under the gradient variance measure CGV (Definition 4.2), with $\hat{n}_m$ satisfying $\frac{\hat{n}_m}{n_m+\hat{n}_m} = \frac{\hat{N}}{N+\hat{N}}$, the objective function $\hat{F}(\theta)$ causes a tighter bounded gradient dissimilarity, i.e., the $CGV(\hat{F}, \theta) = \mathbb{E}_{(x,y)\sim\mathcal{D}_m}||\nabla_{\theta_{low}} f_m(\theta; x, y) - \nabla_{\theta_{low}} F(\theta)||^2 + \frac{N^2}{(N+\hat{N})^2}||\nabla_{\theta_{high}} f_m(\theta; x, y) - \nabla_{\theta_{high}} F(\theta)||^2 \le CGV(F, \theta).$*

*Proof.*

$$CGV(\hat{F}, \theta) = \mathbb{E}_{\substack{(x,y)\sim\mathcal{D}_m\\\hat{h}\sim\mathcal{H}}}||\nabla\hat{f}_m(\theta; x, \hat{h}, y) - \nabla\hat{F}(\theta)||^2$$

$$= \mathbb{E}_{(x,y)\sim\mathcal{D}_m}\left[||\nabla_{\theta_{low}} f_m(\theta; x, y) - \nabla_{\theta_{low}} F(\theta)||^2\right]$$

$$+ \mathbb{E}_{\substack{(x,y)\sim\mathcal{D}_m\\\hat{h}\sim\mathcal{H}}}\left[||\nabla_{\theta_{high}} f_m(\theta; x, y) + \nabla_{\theta_{high}} f_m(\theta; \hat{h}, y) - \nabla_{\theta_{high}}\bar{F}(\theta)||^2\right]. \tag{21}$$

$$\tag{22}$$

On $m$-th client, the number of samples of $(x, y)$ is $n_m$ and the $\hat{h}_m$ is $\hat{n}_m$. Then the high-level gradient variance becomes:

$$\mathbb{E}_{\substack{(x,y)\sim\mathcal{D}_m \\ \hat{h}\sim\mathcal{H}}}[||\frac{n_m}{n_m + \hat{n}_m}\nabla_{\theta_{high}}f_m(\theta; x, y) + \frac{\hat{n}_m}{n_m + \hat{n}_m}\nabla_{\theta_{high}}f_m(\theta; \hat{h}, y) - \nabla_{\theta_{high}}\bar{F}(\theta)||^2$$

$$=\mathbb{E}_{\substack{(x,y)\sim\mathcal{D}_m \\ \hat{h}\sim\mathcal{H}}}[||\frac{n_m}{n_m + \hat{n}_m}\nabla_{\theta_{high}}f_m(\theta; x, y) + \frac{\hat{n}_m}{n_m + \hat{n}_m}\nabla_{\theta_{high}}f_m(\theta; \hat{h}, y)$$

$$- \sum_{m=1}^{M}\frac{n_m + \hat{n}_m}{N + \hat{N}}(\frac{n_m}{n_m + \hat{n}_m}\nabla_{\theta_{high}}f_m(\theta; x, y) + \frac{\hat{n}_m}{n_m + \hat{n}_m}\nabla_{\theta_{high}}f_m(\theta; \hat{h}, y))||^2$$

$$=\mathbb{E}_{(x,y)\sim\mathcal{D}_m}||\frac{n_m}{n_m + \hat{n}_m}\nabla_{\theta_{high}}f_m(\theta; x, y) - \sum_{m=1}^{M}\frac{n_m}{N + \hat{N}}\nabla_{\theta_{high}}f_m(\theta; x, y)||^2$$

$$=\frac{N^2}{(N + \hat{N})^2}\mathbb{E}_{(x,y)\sim\mathcal{D}_m}||\nabla_{\theta_{high}}f_m(\theta; x, y) - \sum_{m=1}^{M}\frac{n_m}{N}\nabla_{\theta_{high}}f_m(\theta; x, y)||^2. \qquad (23)$$

Combining Equation 23 and 21, we obtain

$$\text{CGV}(\hat{F}, \theta) =\mathbb{E}_{(x,y)}||\nabla_{\theta_{low}}f_m(\theta; x, y) - \nabla_{\theta_{low}}F(\theta)||^2$$

$$+ \frac{N^2}{(N + \hat{N})^2}||\nabla_{\theta_{high}}f_m(\theta; x, y) - \nabla_{\theta_{high}}F(\theta)||^2,$$

which completes the proof. $\qquad\square$

For the convergence analysis, there have been many convergence analyses of FedAvg from a gradient dissimilarity viewpoint (Woodworth et al., 2020; Lian et al., 2017; Karimireddy et al., 2020). Specifically, the convergence rate is upper bounded by many factors, among which the gradient dissimilarity plays a crucial role in the bound. In this work, we propose a novel approach inspired by the generalization view to reduce the gradient dissimilarity, we thus provide a tighter bound regarding the convergence rate. This is consistent with our experiments, see Table 2.

## C.4 INTERPRETING AND CONNECTING THEORY WITH ALGORITHMS

We summarize our theory and how it motivates our algorithm as following.

- Our work involves two theorems, i.e., Theorem 4.1 and Theorem 4.2, where Theorem 4.1 motivates the proposed method and Theorem 4.2 indicates another advantage of the proposed method.

- Theorem 4.1 is built upon two definitions, i.e., Definition 3.1 and Definition 4.1. Here, Definition 3.1 measures the model's generalizability on a data distribution from the margin theory, and Definition 4.1 is the conditional Wasserstein distance for two distributions.

- Theorem 4.1 shows that promoting the generalization performance requires constructing similar conditional distributions. This motivates our method, i.e., aiming at making all client models trained on similar conditional distributions to obtain higher generalization performance.

- In our work, inspired by Theorem 4.1, we regard latent features as the data discussed in Theorem 4.1. Accordingly, we can construct similar conditional distributions for the latent features and train models using these similar conditional distributions.

- Theorem 4.2 is built upon Definition 4.2, where gradient dissimilarity in FL is quantitatively measured by Definition 4.2.

- Theorem 4.2 shows that our method can reduce gradient dissimilarity to benefit model convergence.

- For advanced architectures, e.g., a well-trained GFlowNet (Bengio et al., 2023), reducing the distance in the feature space can induce the distance in the data space. However, the property could be hard to maintain for other deep networks, e.g., ResNet.

# D    MORE RELATED WORK

## D.1    ADDRESSING NON-IID PROBLEM IN FL

The convergence and generalization performance of Federated Learning (FL) (McMahan et al., 2017) suffers from the heterogeneous data distribution across all clients (Zhao et al., 2018; Li et al., 2020c; Kairouz et al., 2019). There exists a severe divergence between local objective functions of clients, making local models of FL diverge (Li et al., 2020b; Karimireddy et al., 2020), which is called **client drift**.

Although researchers have designed many new optimization methods to address this problem, it is still an open problem. The performance of federated learning under severe Non-IID data distribution is far behind the centralized training. The previous methods that address Non-IID data problems can be classified into the following directions.

**Model Regularization** focuses on calibrating the local models to restrict them not to be excessively far away from the server model. A number of works (Li et al., 2020b; Acar et al., 2021; Karimireddy et al., 2020) add a regularizer of local-global model difference. FedProx (Li et al., 2020b) adds a penalty of the L2 distance between local models to the server model. SCAFFOLD (Karimireddy et al., 2020) utilizes the history information to correct the local updates of clients. FedDyn (Acar et al., 2021) proposes to dynamically update the risk objective to ensure the device optima is asymptotically consistent. FedIR (Hsu et al., 2020) applies important weight to the client's local objectives to obtain an unbiased estimator of loss. MOON (Li et al., 2021c) adds the local-global contrastive loss to learn a similar representation between clients. CCVR (Luo et al., 2021) transmits the statistics of logits and label information of data samples to calibrate the classifier. FedETF (Li et al., 2023b) proposed a synthetic and fixed ETF classified to resolve the classifier delemma, which is orthogonal to our method. SphereFed (Dong et al., 2022) proposed constraining learned representations of data points to be a unit hypersphere shared by clients. Specifically, in SphereFed, the classifier is flexed with weights spanning the unit hypersphere, and calibrated by a mean squared loss. Besides, SphereFed discovers that the non-overlapped feature distributions for the same class lead to weaker consistency of the local learning targets from another perspective. FedImpro alleviate this problem by estimating and sharing similar features.

**Reducing Gradient Variance** tries to correct the local updates directions of clients via other gradient information. This kind (Wang et al., 2020a; Hsu et al., 2019; Reddi et al., 2021) of methods aims to accelerate and stabilize the convergence. FedNova (Wang et al., 2020a) normalizes the local updates to eliminate the inconsistency between the local and global optimization objective functions. Adjusting data sampling order in local clients is verified to accelerate convergence (Tang et al., 2021). FedAvgM (Hsu et al., 2019) exploits the history updates of the server model to rectify clients' updates. FEDOPT (Reddi et al., 2021) proposes a unified framework of FL. It considers the clients' updates as the gradients in centralized training to generalize the optimization methods in centralized training into FL. FedAdaGrad and FedAdam are FL versions of AdaGrad and Adam. FedSpeed (Sun et al., 2023b) utilized a prox-correction term on local updates to reduce the biases introduced by the prox-term, as a necessary regularizer to maintain the strong local consistency. FedSpeed further merges the vanilla stochastic gradient with a perturbation computed from an extra gradient ascent step in the neighborhood, which can be seen as reducing gradient heterogeneity from another perspective.

**Sharing Features.**    Personalized Federated Learning hopes to make clients optimize different personal models to learn knowledge from other clients and adapt their own datasets (Tan et al., 2022). The knowledge transfer of personalization is mainly implemented by introducing personalized parameters (Liang et al., 2020; Thapa et al., 2020; Li et al., 2021a), or knowledge distillation (He et al., 2020a; Lin et al., 2020; Li & Wang, 2019; Bistritz et al., 2020) on shared local features or extra datasets. Due to the preference for optimizing local objective functions, however, personalized federated models do not have a comparable generic performance (evaluated on global test dataset) to normal FL (Chen & Chao, 2021). Our main goal is to learn a better generic model. Thus, we omit comparisons to personalized FL algorithms.

Except Personalized Federated Learning, some other works propose to share features to improve federated learning. Cronus (Chang et al., 2019) proposes sharing the logits to defend the poisoning attack. CCVR (Luo et al., 2021) transmit the logits statistics of data samples to calibrate the last layer of Federated models. CCVR (Luo et al., 2021) also share the parameters of local feature distribution.

However, we do not need to share the number of different labels with the server, which protects the privacy of label distribution of clients. Moreover, our method acts as a framework for exploiting the sharing features to reduce gradient dissimilarity. The feature estimator does not need to be the Gaussian distribution of local features. One may utilize other estimators or even features of some extra datasets rather than the private ones.

**Sharing Data.** The original cause of client drift is data heterogeneity. Some researchers find that sharing a part of private data can significantly improve the convergence speed and generalization performance (Zhao et al., 2018), yet it sacrifices the privacy of clients' data.

Thus, to both reduce data heterogeneity and protect data privacy, a series of works (Hardt & Rothblum, 2010; Hardt et al., 2012; Chatalic et al., 2021; Johnson et al., 2018; Cai et al., 2021; Li et al., 2023a; Yang et al., 2023) add noise on data to implement sharing data with privacy guarantee to some degree. Some other works focus on sharing a part of synthetic data(Jeong et al., 2018; Long et al., 2021; Goetz & Tewari, 2020; Hao et al., 2021) or data statistics (Shin et al., 2020; Yoon et al., 2021) to help reduce data heterogeneity rather than raw data.

FedDF (Lin et al., 2020) utilizes other data and conducts knowledge distillation based on these data to transfer knowledge of models between server and clients. The core idea of FedDF is to conduct finetuning on the aggregated model via the knowledge distillation with the new shared data.

**Communication compressed FL** Communication compression methods aim to reduce the communication size in each round. Typical methods include **sparsifying** most of unimportant weights (Dorfman et al., 2023; Bibikar et al., 2022; Qiu et al., 2022; Tang et al., 2022a; 2020; 2023b), **quantizing** model updates with fewer bits than the conventional 32 bits (Reisizadeh et al., 2020; Gupta et al., 2023), and **low-rank decomposition** to communicate smaller matrices (Nguyen et al., 2024; Konečný et al., 2016). Despite reducing communication costs, these methods are not as cost-effective as one-shot FL and our methods, due to the necessary of enormous communication rounds for convergence.

## D.2    MEASURING CONTRIBUTION FROM CLIENTS

**Generalization Contribution.** Clients are only willing to participate a FL training when given enough rewards. Thus, it is important to measure their contributions to the model performance (Yu et al., 2020; Ng et al., 2020; Liu et al., 2022; Sim et al., 2020).

There have been some works (Yuan et al., 2022; Yu et al., 2020; Ng et al., 2020; Liu et al., 2022; Sim et al., 2020) proposed to measure the generalization contribution from clients in FL. Some works (Yuan et al., 2022) propose to experimentally measure the performance gaps from the unseen client distributions. Data shapley (Ghorbani & Zou, 2019; Yu et al., 2020; Luo et al., 2023; Liu et al., 2023) is proposed to measure the generalization performance gain of client participation. (Liu et al., 2022) improves the calculation efficiency of Data Shapley. And there is some other work that proposes to measure the contribution by learning-based methods (Zhan et al., 2020). Our proposed questions are different from these works. Precisely, these works measure the generalization performance gap with or without some clients that never join the collaborative training of clients. However, we hope to understand the contribution of clients at each communication round. Based on this understanding, we can further improve the FL training and obtain a better generalization performance.

It has been empirically verified that a large number of selected clients introduces new challenges to optimization and generalization of FL (Charles et al., 2021), although some theoretical works show the benefits from it (Yang et al., 2020). This encourages us to understand what happens during the local training and aggregation. Causality is also a potential way to explore the client contribution (Zhang et al., 2021).

**Client Selection.** Several works (Cho et al., 2020; Goetz et al., 2019; Ribero & Vikalo, 2020; Lai et al., 2021) propose new algorithms to strategically select clients rather than randomly. However, these methods only consider the hardware resources or local generalization ability. How local training affects the global generalization ability has not been explored.

## D.3    SPLIT TRAINING

To efficiently train neural networks, split training instead of end-to-end training is proposed to break the forward, backward, or model updating dependency between layers of neural networks.

Table 4: Demystifying different FL algorithms related to the sharing data and features.

| | Shared Thing | Low-level Model | Objective |
|---|---|---|---|
| (Chatalic et al., 2021; Cai et al., 2021) | Raw Data With Noise | Shared | Others |
| (Long et al., 2021; Hao et al., 2021) | Params. of Data Generator | Shared | Global Model Performance |
| (Yoon et al., 2021; Shin et al., 2020) | STAT. of raw Data | Shared | Global Model Performance |
| (Luo et al., 2021) | STAT. of Logis, Label Distribution | Shared | Global Model Performance |
| (Chang et al., 2019) | Hidden Features | Shared | Defend Poisoning Attack |
| (Li & Wang, 2019; Bistritz et al., 2020) | logits | Private | Personalized FL |
| (He et al., 2020a; Liang et al., 2020) | Hidden Features | Private | Personalized FL |
| (Thapa et al., 2020; Oh et al., 2022) | Hidden Features | Shared | Accelerate Training |
| **FedImpro** | Params. of Estimated Feat. Distribution | Shared | Global Model Performance |

Note: "STAT." means statistic information, like mean or standard deviation, "Feat." means hidden features, "Params." means parameters.

To break the backward dependency on subsequent layers, hidden features could be forwarded to another loss function to obtain the **Local Error Signals** (Marquez et al., 2018; Nøkland & Eidnes, 2019; Löwe et al., 2019; Wang et al., 2020b; Zhuang et al., 2021). How to design a suitable local error still remains as an open problem. Some works propose to utilize extra modules to synthesize gradients (Jaderberg et al., 2017), so that the backward and updates of different layers can be decoupled. **Features Replay** (Huo et al., 2018) is to reload the history features of the preceding layers into the next layers. By reusing the history features, the calculation on different layers could be asynchronously conducted.

Some works propose Split FL (SFL) to utilize split training to accelerate federated learning (Oh et al., 2022; Thapa et al., 2020). In SFL, the model is split into client-side and server-side parts. At each communication round, the client only downloads the client-side model from the server, conducts forward propagation, and sends the hidden features to the server for computing loss and backward propagation. This method aims to accelerate FL's training speed on the client side and cannot support local updates. In addition, sending all raw features could introduce a high data privacy risk. Thus, we omit the comparisons to these methods.

We demystify different FL algorithms related to the shared features in Table 4.

## E    DETAILS OF EXPERIMENT CONFIGURATION

### E.1    HARDWARE AND SOFTWARE CONFIGURATION

**Hardware and Library.** We conduct experiments using GPU GTX-2080 Ti, CPU Intel(R) Xeon(R) Gold 5115 CPU @ 2.40GHz. The operating system is Ubuntu 16.04.6 LTS. The Pytorch version is 1.8.1. The Cuda version is 10.2.

**Framework.** The experiment framework is built upon a popular FL framework FedML (He et al., 2020b; Tang et al., 2023a). And we conduct the standalone simulation to simulate FL. Specifically, the computation of client training is conducted sequentially, using only one GPU, which is a mainstream experimental design of FL (He et al., 2020b; Sun et al., 2023b; Reddi et al., 2021; Luo et al., 2021; Liang et al., 2020; Li et al., 2021b; Caldas et al., 2018), due to it is friendly to simulate large number of clients. Local models are offloaded to CPU when it is not being simulated. Thus, the real-world computation time would be much less than the reported computation time. Besides, the communication does not happen, which would be the main bottleneck in real-world FL.

### E.2    HYPER-PARAMETERS

The learning rate configuration has been listed in Table 5. We report the best results and their learning rates (grid search in $\{0.0001, 0.001, 0.01, 0.1, 0.3\}$).

And for all experiments, we use SGD as optimizer for all experiments, with batch size of 128 and weight decay of 0.0001. Note that we set momentum as 0 for baselines, as we find the momentum of 0.9 may harm the convergence and performance of FedAvg in severe Non-IID situations. We also report the best test accuracy of baselines that are trained with momentum of 0.9 in Table 6. The client-side momentum in FL training does not always commit better convergence because the momentum introduces larger local updates, increasing the client drift, which is also observed in a

recent benchmark (He et al., 2021). And the server-side momentum (Hsu et al., 2019) may improve the performance. The compared algorithms including FedAvg, FedProx, SCAFFOLD, FedNova do not use the server-side momentum. For the fair comparisons we did not use the server-side momentum for all algorithms.

For $K = 10$ and $K = 100$, the maximum communication round is 1000, For $K = 10$ and $E = 5$, the maximum communication round is 400 (due to the $E = 5$ increase the calculation cost). The number of clients selected for calculation is 5 per round for $K = 10$, and 10 for $K = 100$.

Table 5: Learning rate of all experiments.

| Dataset | FL Setting | | | FedAvg | FedProx | SCAFFOLD | FedNova | **FedImpro** |
|---|---|---|---|---|---|---|---|---|
| | $a$ | $E$ | $K$ | | | | | |
| CIFAR-10 | 0.1 | 1 | 10 | 0.1 | 0.1 | 0.1 | 0.1 | 0.1 |
| | 0.05 | 1 | 10 | 0.1 | 0.1 | 0.01 | 0.1 | 0.1 |
| | 0.1 | 5 | 10 | 0.1 | 0.1 | 0.1 | 0.1 | 0.1 |
| | 0.1 | 1 | 100 | 0.1 | 0.1 | 0.01 | 0.1 | 0.1 |
| FMNIST | 0.1 | 1 | 10 | 0.1 | 0.1 | 0.1 | 0.1 | 0.1 |
| | 0.05 | 1 | 10 | 0.1 | 0.1 | 0.001 | 0.1 | 0.1 |
| | 0.1 | 5 | 10 | 0.1 | 0.1 | 0.1 | 0.1 | 0.1 |
| | 0.1 | 1 | 100 | 0.1 | 0.1 | 0.01 | 0.1 | 0.1 |
| SVHN | 0.1 | 1 | 10 | 0.1 | 0.1 | 0.01 | 0.1 | 0.1 |
| | 0.05 | 1 | 10 | 0.1 | 0.1 | 0.01 | 0.1 | 0.1 |
| | 0.1 | 5 | 10 | 0.1 | 0.01 | 0.01 | 0.01 | 0.1 |
| | 0.1 | 1 | 100 | 0.1 | 0.1 | 0.001 | 0.1 | 0.1 |
| CIFAR-100 | 0.1 | 1 | 10 | 0.1 | 0.1 | 0.1 | 0.1 | 0.1 |
| | 0.05 | 1 | 10 | 0.1 | 0.1 | 0.1 | 0.1 | 0.1 |
| | 0.1 | 5 | 10 | 0.1 | 0.1 | 0.1 | 0.1 | 0.1 |
| | 0.1 | 1 | 100 | 0.1 | 0.1 | 0.1 | 0.1 | 0.1 |

### E.3 MORE CONVERGENCE FIGURES

Except the Figure 2 in the main paper, we provide more convergence results as Figures 4, 5, 6 and 7. These results show that our method can accelerate FL training and obtain higher generalization performance.

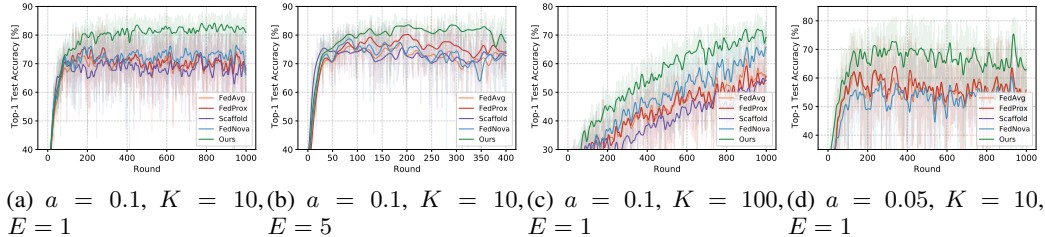

(a) $a = 0.1$, $K = 10$, $E = 1$ (b) $a = 0.1$, $K = 10$, $E = 5$ (c) $a = 0.1$, $K = 100$, $E = 1$ (d) $a = 0.05$, $K = 10$, $E = 1$

Figure 4: Convergence comparison of CIFAR-10.

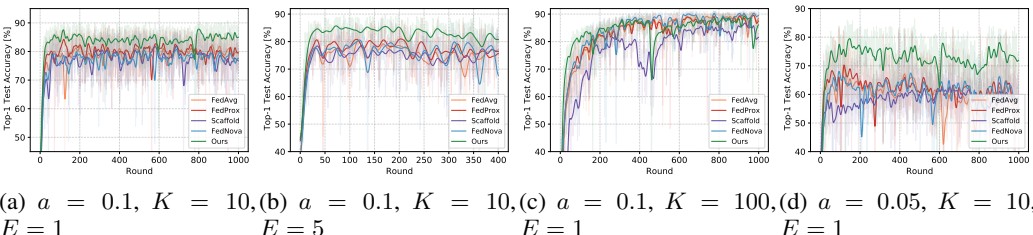

(a) $a = 0.1$, $K = 10$, $E = 1$ (b) $a = 0.1$, $K = 10$, $E = 5$ (c) $a = 0.1$, $K = 100$, $E = 1$ (d) $a = 0.05$, $K = 10$, $E = 1$

Figure 5: Convergence comparison of FMNIST.

Table 6: Baselines with Momentum-SGD.

| Dataset | FL Setting | | | FedAvg | FedProx | SCAFFOLD | FedNova |
| --- | --- | --- | --- | --- | --- | --- | --- |
| | $a$ | $E$ | $K$ | | | | |
| CIFAR-10 | 0.1 | 1 | 10 | 79.98 | 83.56 | 83.58 | 81.35 |
| | 0.05 | 1 | 10 | 69.02 | 78.66 | 38.55 | 64.78 |
| | 0.1 | 5 | 10 | 84.79 | 82,18 | 86.20 | 86.09 |
| | 0.1 | 1 | 100 | 49.61 | 49.97 | 52.24 | 46.53 |
| FMNIST | 0.1 | 1 | 10 | 86.81 | 87.12 | 86.21 | 86.99 |
| | 0.05 | 1 | 10 | 78.57 | 81.96 | 76.08 | 79.06 |
| | 0.1 | 5 | 10 | 87.45 | 86.07 | 87.10 | 87.53 |
| | 0.1 | 1 | 100 | 90.11 | 90.71 | 85.99 | 87.09 |
| SVHN | 0.1 | 1 | 10 | 88.56 | 86.51 | 80.61 | 89.12 |
| | 0.05 | 1 | 10 | 82.67 | 78.57 | 74.23 | 82.22 |
| | 0.1 | 5 | 10 | 87.92 | 78.43 | 81.07 | 88.17 |
| | 0.1 | 1 | 100 | 89.44 | 89.51 | 89.55 | 82.08 |
| CIFAR-100 | 0.1 | 1 | 10 | 67.95 | 65.29 | 67.14 | 68.26 |
| | 0.05 | 1 | 10 | 62.07 | 61.52 | 59.04 | 60.35 |
| | 0.1 | 5 | 10 | 69.81 | 62.62 | 70.68 | 70.05 |
| | 0.1 | 1 | 100 | 48.33 | 48.14 | 51.63 | 48.12 |

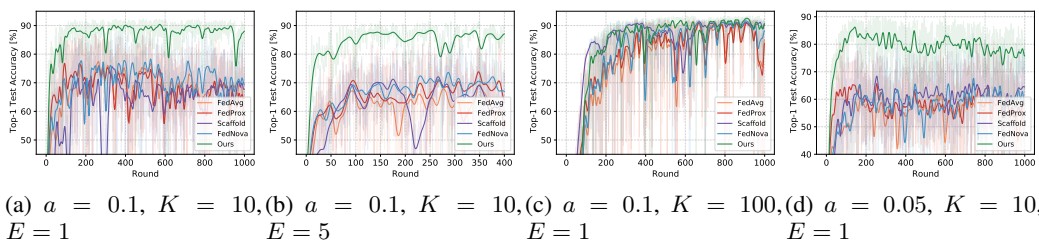

(a) $a = 0.1$, $K = 10$, $E = 1$  (b) $a = 0.1$, $K = 10$, $E = 5$  (c) $a = 0.1$, $K = 100$, $E = 1$  (d) $a = 0.05$, $K = 10$, $E = 1$

Figure 6: Convergence comparison of SVHN.

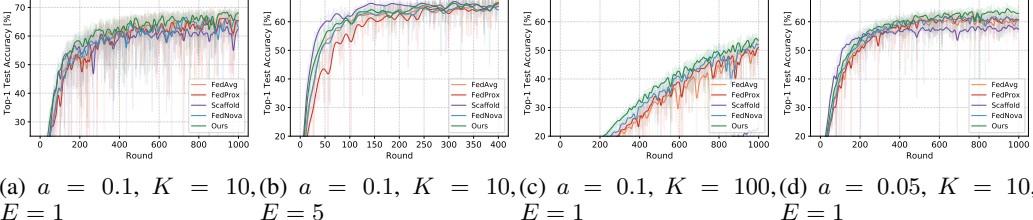

(a) $a = 0.1$, $K = 10$, $E = 1$  (b) $a = 0.1$, $K = 10$, $E = 5$  (c) $a = 0.1$, $K = 100$, $E = 1$  (d) $a = 0.05$, $K = 10$, $E = 1$

Figure 7: Convergence comparison of CIFAR100.

## F ADDITIONAL EXPERIMENTS

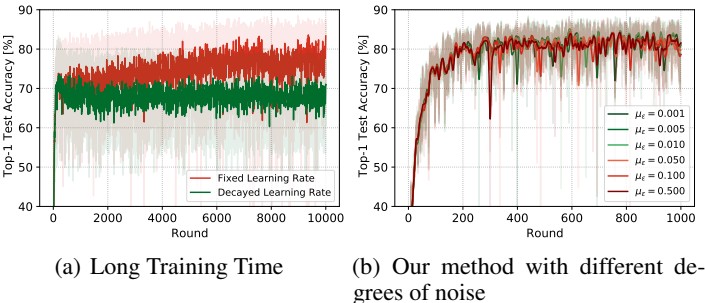

(a) Long Training Time  (b) Our method with different degrees of noise

Figure 8: Additional experiments on CIFAR-10 with $a = 0.1$, $K = 10$, $E = 1$.

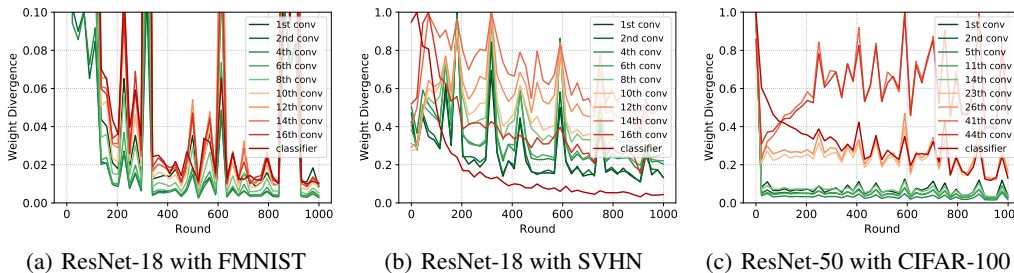

| (a) ResNet-18 with FMNIST | (b) ResNet-18 with SVHN | (c) ResNet-50 with CIFAR-100 |

Figure 9: Layer divergence of FedAvg.

Table 7: Test Accuracy of our method with different degrees of noise.

| $\mu_\epsilon$ | 0.0 | 0.001 | 0.005 | 0.01 | 0.05 | 0.1 | 0.5 |
|---|---|---|---|---|---|---|---|
| Test Accuracy (%) | 88.45 | 88.43 | 88.23 | 88.26 | 88.07 | 88.11 | 88.3 |

Table 8: Test Accuracy of different algorithms on FEMNIST.

| | FedAvg | FedProx | FedNova | **FedImpro** |
|---|---|---|---|---|
| Test Accuracy (%) | 80.83 | 79.70 | 68.96 | **82.77** |
| Comm. Round to attain the Target Acc. | 82 | NaN | NaN | **45** |

## F.1 TRAINING WITH LONGER TIME

To demonstrate the difficulty of optimization of FedAvg in heterogeneous-data environment, we show the results of training 10000 rounds, as shown in Figure 8 (a). During this 10000 rounds, the highest test accuracy of FedAvg with fixed learning rate is 88.5%, and it of the FedAvg with decayed learnign rate is 82.65%. Note that we set the learning rate decay exponentially decay at each communication round, wich rate 0.997. Even after 2000 rounds, the learning rate becomes as the around 0.0026 times as the original learning rate. The results show that the longer training time cannot fill the generalization performance gap between FedAvg and centralized training, encouraging us to develop new optimization schemes to improve it.

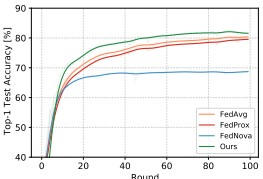

Figure 10: Convergence comparison of FEMNIST with 3400 clients.

## F.2 SHARING ESTIMATING PARAMETERS WITH NOISES OF DIFFERENT DEGREES

To enhance the security of the sharing feature distribution, we add the noise $\epsilon \sim \mathcal{N}(0, \sigma\epsilon)$ on the $\sigma_m$ and $\mu_m$. The privacy degree could be enhanced by the larger $\mu_\epsilon$. We show the results of our method with different $\sigma_\epsilon$ in Figure 8 (b) and Table 7. The results show that under the high perturbation of the estimated parameters, our method attains both high privacy and generalization gains.

## F.3 MORE EXPERIMENTS OF THE REAL-WORLD DATASETS

To verify the effect of our methods on the real-world FL datasets, we conduct experiments with Federated EMNIST(FEMNIST) (Caldas et al., 2018; He et al., 2020b), which has 3400 users, 671585 training samples and 77483 testing samples. We sample 20 clients per round, and conduct local training with 10 epochs. We search the learning rate for algorithms in $\{0.01, 0.05, 0.1\}$ and find the 0.05 is the best for all algorithm. Figure 10 and Table 8 show that our method converges faster

and attains better generalization performance than other methods. Note that the SCAFFOLD is not included the experiments, as it has a very high requirement (storing the control variates) of simulating 3400 clients with few machines.

**More Results of the Layer-wise Divergence.** We conduct more experiments of the layer divergence of FedAvg with different datasets including FMNIST, SVHN and CIFAR-100, training with ResNet-18 and ResNet-50. As Figure 3 and 9 shows, the divergence of the low-level model divergence shrinks faster than the high-level. Thus, reducing the high-level gradient dissimilarity is more crucial than the low-level.

## F.4 MORE RESULTS OF DIFFERENT MODEL ARCHITECTURE

To verify the effect of our method on different model architectures, we conduct additional experiments of training VGG-9 on CIFAR-10, instead of the ResNet architectures. The experimental results are shown in Table 9. The target accuracy is 82%. We can see that our method can outperform baseline methods with different architectures.

Table 9: Test Accuracy of different algorithms with VGG-9 on CIFAR-10.

|  | FedAvg | FedProx | SCAFFOLD | FedNova | **FedImpro** |
|---|---|---|---|---|---|
| Test Accuracy (%) | 82.58 | 82.92 | 82.43 | 82.91 | **84.52** |
| Comm. Round to attain the Target Acc. | 836 | 844 | 512 | 664 | **426** |

## F.5 MORE BASELINES

We further compare our method with FedDyn (Acar et al., 2021), CCVR (Luo et al., 2021), Fed-Speed (Sun et al., 2023b) and FedDF (Lin et al., 2020). The results are given in Table 10 as below. As original experiments in these works do not utilize the same FL setting, we report their test accuracy of CIFAR-10/100 dataset and the according settings in original papers. To make the comparison fair, we choose the same or the harder settings of their methods and report their original results. The larger dirichlet parameter $\alpha$ means more severe data heterogeneity. The higher communication round means training with longer time. We can see that our method can outperform these methods with the same setting or even the more difficult settings for us.

Table 10: Comparisons with more baselines.

| Method | Dataset | Dirichlet $\alpha$ | Client Number | Communication Round | Test ACC. |
|---|---|---|---|---|---|
| FedDyn | | 0.3 | 100 | 1400 | 77.33 |
| FedSpeed | CIFAR-10 | 0.3 | 100 | 1400 | 82.68 |
| FedImpro | | 0.3 | 100 | 1400 | **85.14** |
| FedImpro | | 0.1 | 100 | 1400 | **82.76** |
| CCVR | | 0.1 | 10 | 100 | 62.68 |
| FedDF (no extra data) | CIFAR-10 | 0.1 | 10 | 100 | 38.6 |
| FedImpro | | 0.1 | 10 | 100 | **68.29** |
| SphereFed | CIFAR-100 | 0.1 | 10 | 100 | 69.19 |
| FedImpro | | 0.1 | 10 | 100 | **70.28** |

## F.6 MORE ABLATION STUDY ON NUMBER OF SELECTED CLIENTS

The feature distribution depends on the client selected in each round. Thus, to analyze the number of clients on the feature distribution estimation, we conduct ablation study with training CIFAR-10 datasets with varying the number of selected clients. Table 11 shows the effect of varying the number of selected clients per round. More clients can improve both FedAvg and FedImpro. The FedImpro can work well with more selected clients.

## F.7 RECONSTRUCTED RAW DATA BY MODEL INVERSION ATTACKS

We utilize the model inversion method (Zhao et al., 2021; Zhou et al., 2023) used in (Luo et al., 2021) to verify the privacy protection of our methods. The original private images are shown in Figure 11,

| $a$ | E | $M$ | $M_c$ | FedAvg | FedImpro |
|-----|---|-----|-------|--------|----------|
| 0.1 | 1 | 10 | 5 | 83.65±2.03 | **88.45±0.43** |
| 0.1 | 1 | 10 | 10 | 86.65±1.26 | **90.25±0.93** |
| 0.05 | 1 | 10 | 5 | 75.36±1.92 | **81.75±1.03** |
| 0.05 | 1 | 10 | 10 | 78.36±1.58 | **85.75±1.21** |
| 0.1 | 5 | 10 | 5 | 85.69±0.57 | **88.10±0.20** |
| 0.1 | 5 | 10 | 10 | 87.92±0.31 | **90.92±0.25** |
| 0.1 | 1 | 100 | 10 | 73.42±1.19 | **77.56±1.02** |
| 0.1 | 1 | 100 | 20 | 77.82±0.94 | **85.39±1.15** |

Table 11: Ablation study with training on CIFAR-10 on different number of selected clients.

reconstructed images using the features of each image are shown in Figure 12, reconstructed images using the mean of features of all images are shown in Figure 13. Based on the results, we observe that feature inversion attacks struggle to reconstruct private data using the shared feature distribution parameters $\sigma_m$ and $\mu_m$. Thus, our method is robust against model inversion attacks.

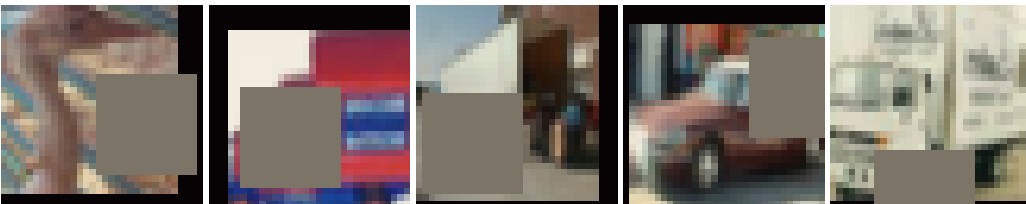

Figure 11: Original private images that are fed into the model.

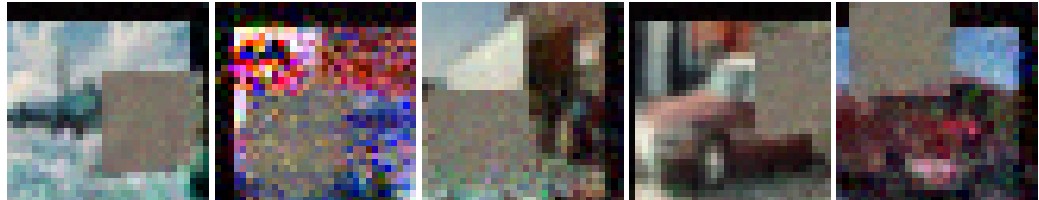

Figure 12: Reconstructed images using the raw features. Note that each image is reconstructed by the feature of each private image, i.e. sharing the raw features. It shows that the feature inversion method can successfully reconstruct the original image based on the raw features.

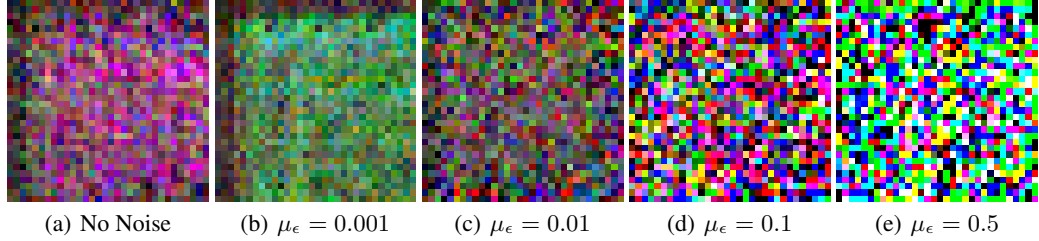

(a) No Noise      (b) $\mu_\epsilon = 0.001$      (c) $\mu_\epsilon = 0.01$      (d) $\mu_\epsilon = 0.1$      (e) $\mu_\epsilon = 0.5$

Figure 13: Reconstructed images using the mean of features (**our methods**) of all images with noises of different degrees. $\mu_\epsilon$ is the variance of the Gaussian noise. Now, feature inversion method cannot reconstruct the original images..

