# OpenReview forum: "FedImpro: Measuring and Improving Client Update in Federated Learning"
_ICLR.cc/2024/Conference — ICLR 2024 poster_

### Official Review · Reviewer_LhTp · 2023-10-31

**Soundness:** 3 good
**Presentation:** 3 good
**Contribution:** 3 good
**Rating:** 6
**Confidence:** 4

**Summary:**

The paper proposes FedImpro to mitigate the client drift problem in FL. It provides a new way to correct gradients and updates. The method mainly has 2 parts. The first is the study of the generalization contribution of each client, then proposes to leverage the same or similar conditional distributions for local training. The second is to propose decoupling a deep neural network into a low-level model (features extraction) and a high-level model (classifier network).

**Strengths:**

Even though decoupling a neural network into a feature extractor network and a classifier network is not novel, the paper proposed to combine the decoupling method with the features distribution estimation method with privacy protection is quite novel for mitigating client drift. Also, the sampling part with the synthetic features to increase the distribution similarity is quite interesting too.

**Weaknesses:**

- All the distributions in the theoretical analysis considered conditional distribution conditioned on the label y, and then the paper said that ' it is straightforward to make all client models trained on similar distributions to obtain higher generalization performance'. But, for a dataset such as CIFAR10, when we partition it among clients, the non-IID is introduced by the label imbalanced across clients, which means that the conditional distribution on the label is the same. However, we would still experience client drift in this case. I think more explanation/analysis is required on this aspect. Also, since the analysis is on the conditional distribution, the experiments should reflect that, by using a dataset with the natural partition, such as FEMNIST.

- since the feature distribution depends on the client selected in each round. Unlike the standard FedAvg training, the number of clients per round only impacts the convergence rate. Here for the FedImprov method, it will also impact the distribution estimation. Therefore, an ablation study analyzing the number of clients on the feature distribution estimation is lacking.

Minor:
- a small typo in Sec 5.1, M=10 should be cross-silo and M=100 should be cross-device.

**Questions:**

- do you use h_hat to update the low-level model?

---

> ### Author Response · Authors · 2023-11-16
> **Response to Reviewer LhTp [1/2]**
>
> We sincerely thank the reviewer for taking the time to review. We appreciate that you find that estimation method with privacy protection is quite novel, and sampling part with the synthetic features to increase the distribution similarity is quite interesting. According to your insightful comments, we provide detailed feedback below. We also add them into our main text or appendix in the revision. We hope these changes and our revision can address your concerns and improve our work.
>
>
> **Q1** More explanation on client drift:
> >  But, for a dataset such as CIFAR10, when we partition it among clients, the non-IID is introduced by the label imbalanced across clients, which means that the conditional distribution on the label is the same. However, we would still experience client drift in this case. I think more explanation/analysis is required on this aspect.
>
> ***Ans for Q1):*** Thanks for your insightful comments. We provide detailed feedbacks as below.
> - **Case 1: The global label distribution is balanced, but local label distribution is different on different clients.** Although the overall label distribution is balanced, one client may have completely different conditional distribution on the label from other clients. Our analysis focuses on this case, and closes the conditional Wasserstein distance between these two clients.
> - **Case 2: The global label distribution is balanced, and local label distribution is also same with different clients.** In this case, the client drift may still happen due to the feature distribution is different [1,2]. In this case, we may utilize the Wasserstein distance not conditioned on label in unsupervised domain adaptation [3,4] to analyse this problem. We will consider this case as the future work.
>
> Thanks again for your constructive comments. We have added these discussion in Appendix B to enhance the quality of our paper.
>
>
> **Q2** Experiment issues:
> >  since the analysis is on the conditional distribution, the experiments should reflect that, by using a dataset with the natural partition, such as FEMNIST
>
> ***Ans for Q2):***
> Thanks for your constructive comments. We conduct experiments with FEMNIST using its natural partition. New results are summarized in Table 4-1, showing that FedImpro can improve the model performance than other baselines.
>
> **Table 4-1**
> |                              | FedAvg | FedProx | FedNova | **FedImpro** |
> |------------------------------|--------|---------|---------|--------------|
> | Test Accuracy (%)            | 80.83  | 79.70   | 68.96   | **82.77**    |
> | Comm. Round to attain Target Acc. | 82     | NaN     | NaN     | **45**       |

---

> ### Author Response · Authors · 2023-11-16
> **Response to Reviewer LhTp [2/2]**
>
> **Q3** Ablation study on the number of clients:
> >  since the feature distribution depends on the client selected in each round. Unlike the standard FedAvg training, the number of clients per round only impacts the convergence rate. Here for the FedImprov method, it will also impact the distribution estimation. Therefore, an ablation study analyzing the number of clients on the feature distribution estimation is lacking.
>
> ***Ans for Q3):***
> Thanks for this constructive comments. We conduct more ablation studies with training CIFAR-10 and varying the number of selected clients per round. The result are summarized in Table 4-2, in which $a$ is the dirichlet parameter, E the local epochs, M the number of total clients, $M_c$ the number of selected clients per round. Results show that increasing the selected clients helps increase the performance of both FedAvg and FedImpro. It means that the feature estimation works well with more selected clients. We have added these results in Appendix of our revision.
>
> **Table 4-2**
> | $a$     | E   | $M$    | $M_c$ |FedAvg| FedImpro       |
> |------|-------|-----|--|----|------|
> |  0.1   | 1   | 10   | 5  | 83.65±2.03     |**88.45±0.43** |
> | 0.1   | 1   | 10   | 10  | 86.65±1.26     | **90.25±0.93** |
> | 0.05  | 1   | 10   | 5  | 75.36±1.92     |**81.75±1.03** |
> | 0.05  | 1   | 10   | 10  | 78.36±1.58     |**85.75±1.21** |
> | 0.1   | 5   | 10   | 5  | 85.69±0.57     |**88.10±0.20** |
> | 0.1   | 5   | 10   | 10  | 87.92±0.31     |**90.92±0.25** |
> | 0.1   | 1   | 100   | 10 | 73.42±1.19    |**77.56±1.02** |
> | 0.1   | 1   | 100   | 20 | 77.82±0.94    |**85.39±1.15** |
>
>
> **Q4** Typos
> >  a small typo in Sec 5.1, M=10 should be cross-silo and M=100 should be cross-device.
>
> ***Ans for Q4):*** Thanks for your careful reading and kind comments.  We have modified the typo and update the text in the revision.
>
>
> **Q5** Interpreting $\hat{h}$
> >  do you use h_hat to update the low-level model?
>
> ***Ans for Q5):*** Thanks for your insightful question. The low-level model will not receive gradients calculated from the $\hat{h}$. Thus, the low-level model is not explicitly updated by the $\hat{h}$. However, from another aspect, training the high-level model benefits from the closer-distance features and the reduced gradient heterogeneity. In FedImpro, the backward-propagated gradients on original features $h$ have lower bias than FedAvg. Therefore, **low-level model can implicitly benefit from $\hat{h}$**. We have added these discussion in Appendix B.
>
>
>
> > ***Reference***
> >
> > [1] Federated Learning on Non-IID Data Silos: An Experimental Study. In ICDE 2023.
> >
> > [2] Advances and Open Problems in Federated Learning. In Foundations and Trends® in Machine Learning 2021.
> >
> > [3] Sliced Wasserstein Discrepancy for Unsupervised Domain Adaptation. In CVPR 2019.
> >
> > [4] Enhanced Transport Distance for Unsupervised Domain Adaptation. In CVPR 2020.
> >

---

> ### Author Response · Authors · 2023-11-18
> **Welcome for more discussions**
>
> Dear reviewer #LhTp,
>
> Thanks for your valuable time in reviewing and constructive comments. Following your comments, we have tried our best to provide responses and revise our paper. Here is a **summary of our response** for your convenience:
> - (1) **Explanation of client drift in another case**: We have provided an explanation about the client drift in cases that conditional distribution on the label is the same.
> - (2) **More experiment on FEMNIST**: We also provided the more experiment results of FEMNIST according to the insightful suggestions. Table 4-1 in our response shows that FedImpro works well with the FEMNIST that is naturally partitioned.
> - (3) **More ablation study**: Following your insightful comments, we conducted the ablation study of different number of selected clients. Results have been added in Table 4-2 in the responses and the revision, showing that increasing the selected clients helps increase the performance of both FedAvg and FedImpro.
> - (4) **Interpreting $\hat{h}$**: According to your insightful question, we have provided a detailed discussion on how $\hat{h}$ influences the low-level model. Simply speaking, it is not explicitly used to update the low-level model, but it can implicitly benefit low-level model training.
>
> We humbly hope our repsonse has addressed your concerns. If you have any additional concerns or comments that we may have missed in our responses, we would be most grateful for any further feedback from you to help us further enhance our work.
>
>
> Best regards
>
> Authors of #4656

---

> ### Author Response · Authors · 2023-11-20
> **Window for responsing and draft updating is closing**
>
> Dear Reviewer #LhTp,
>
> Thanks a lot for your time in reviewing and reading our response and the revision. Thanks very much for your valuable comments. We sincerely understand you’re busy. But as the window for responsing and paper revision is closing, would you mind checking our response ([a brief summary](https://openreview.net/forum?id=giU9fYGTND&noteId=o5r60kxvby), and [details](https://openreview.net/forum?id=giU9fYGTND&noteId=oRG5OUQqhx)) and confirm whether you have any further questions? We are looking forward to your reply.
>
> Best regards and thanks,
>
> Authors of #4656

---

> > ### Comment · Reviewer_LhTp · 2023-11-20
> >
> > Thanks a lot for the additional work. I have some more questions regarding the response:
> > - Appendix B: without mentioning the exact setup/framework/hardware, only quoting walk-clock time seems meaningless in this scenarios. I would suggest to add the specific setup/framework/hardware for reference there.
> > - I still think the explanation for h_{hat} is a bit lacking. Even though the authors mention that it can implicitly benefit low-level model training, the argument without quantitative results seems a bit weak here.
> > - Also, do you fedavg the low-level model or keep the low-level model locally for each clients?

---

> > > ### Author Response · Authors · 2023-11-21
> > > **Thanks for your swift reply and new comments [1/2]**
> > >
> > > Dear Reviewer #LhTp,
> > >
> > > Thanks for your swift reply and new insightful questions despite such a busy period. Please see our following feedbacks based on your new questions.
> > >
> > >
> > > **Q1:** Experiment setup/framework/hardware.
> > > > Appendix B: without mentioning the exact setup/framework/hardware, only quoting walk-clock time seems meaningless in this scenarios. I would suggest to add the specific setup/framework/hardware for reference there.
> > >
> > > ***Ans for Q1:***
> > > Thanks for your construtive comments. Please see our added descriptions of experiment hardware and software, which are also added in Appendix E.1. And we add following setup illustration in Appendix B.
> > > - **Hardware and Library.** We conduct experiments using GPU GTX-2080 Ti, CPU Intel(R) Xeon(R) Gold 5115 CPU @ 2.40GHz. The operating system is Ubuntu 16.04.6 LTS. The Pytorch version is 1.8.1. The Cuda version is 10.2.
> > > - **Framework.** The experiment framework is built upon a popular FL framework FedML [1]. And we conduct the standalone simulation to simulate FL. Specifically, the **computation of client training is conducted sequentially, using only one GPU**, which is a mainstream experimental design of FL [1-8] due to it is friendly to simulate large number of clients. Local models are offloaded to CPU when it is not being simulated. Thus, **the real-world computation time would be much less than the reported computation time**. Besides, the communication does not happen, which would be the main bottleneck in real-world FL.
> > > - **Setup.** The reported running time in Appendix B is based on this test setup: Training ResNet-18 on Dirichlet partitioned CIFAR-10 dataset with  $\alpha=0.1$, $M=10$, sampling 5 clients each round, with total communication rounds as 1000 and local epoch as 1.

---

> > > ### Author Response · Authors · 2023-11-21
> > > **Thanks for your swift reply and new comments [2/2]**
> > >
> > > **Q2:** More explanation and quantitative results for $\hat{h}$.
> > > > I still think the explanation for h_{hat} is a bit lacking. Even though the authors mention that it can implicitly benefit low-level model training, the argument without quantitative results seems a bit weak here.
> > >
> > > ***Ans for Q2:*** Thanks for your constructive comments.
> > > - To provide a quantitative analysis of how $\hat{h}$ influences low-level model, we need to conduct an ablation study by removing the benefits on high-level model from $\hat{h}$. However, this is non-trivial. The implicit benefits, i.e. better backward gradients from the high-level models, require the high-level models are also better ("better" means that they are good for model generalization and defending data heterogeneity). Thus, the benefits on high-level and low-level model are coupled. Thus, how to build an explicit bridge to benefit training low-level model from $\hat{h}$ is still under exploring.
> > > - Here, we provide a possible way for future works to improve low-level model through some design of $\hat{h}$. Specifically, we can design a method to let the high-level model selects out those values in $\hat{h}$ that are invariant feature [13,14], and encourages low-level models to extract these features, which are generic and cross-domain.
> > >
> > >
> > > **Q3:** Explanation about the low-level models.
> > > > Also, do you fedavg the low-level model or keep the low-level model locally for each clients?
> > >
> > > ***Ans for Q3:*** Thank you for this insightful question, which inspire us more interesting ideas. We provide the following feedbacks and add them into discussion in Appendix B.
> > > - FedImpro focuses on how to better learning a global model instead of personalized or heterogeneous models. Thus, we conduct **fedavg on the low-level model**.
> > > - The core idea of FedImpro, sharing estimated features would be very helpful in personalized split-FL [9,10,11,12] (*keep the low-level model locally*), with these reasons: (a) our **Theorem 4.1 and 4.2 can also be applied into personalized FL with sharing estimated features**; (b) The large communication overhead in personalized split-FL is large due to the communication in each forward and backward propagation (low-level and high-level models are typically deployed in clients and server, respectively). Now, **sharing estimated features would help reduce this communication a lot by decoupling the forward and backward propagations**. Specifically, low-level model can focus on learning a more accurate feature estimation and share it with high-level model. High-level models can repeatedly sample features from the estimated distributions for training, and periodically (not each iteration) send gradients to clients to learn a better feature extractor.
> > >
> > > We are looking forward to your reply and happy to answer your further questions.
> > >
> > > Best regards and thanks,
> > >
> > > Authors of #4656
> > >
> > >
> > > > ***Reference***
> > > > [1] FedML: A Research Library and Benchmark for Federated Machine Learning. In Arxiv 2020.
> > > >
> > > > [2] Sun, Yan, et al. "Fedspeed: Larger local interval, less communication round, and higher generalization accuracy." ICLR (2023).
> > > >
> > > > [3] Adaptive Federated Optimization. In ICLR 2021.
> > > >
> > > > [4] FedScale: Benchmarking Model and System Performance of Federated Learning at Scale. In ICML 2022.
> > > >
> > > > [5] No Fear of Heterogeneity: Classifier Calibration for Federated Learning with Non-IID Data. In NeurIPS 2021.
> > > >
> > > > [6] Think locally, act globally: Federated learning with local and global representations. In Arxiv 2020.
> > > >
> > > > [7] Federated Learning on Non-IID Data Silos: An Experimental Study. In ICDE 2023.
> > > >
> > > > [8] LEAF: A Benchmark for Federated Settings. In Arxiv 2018.
> > > >
> > > > [9] Think locally, act globally: Federated learning with local and global representations. In Ariv 2020.
> > > >
> > > > [10] Exploiting Shared Representations for Personalized Federated Learning. In ICML 2021.
> > > >
> > > > [11] Splitfed: When federated learning meets split learning. In Arxiv 2020.
> > > >
> > > > [12] On Bridging Generic and Personalized Federated Learning for Image Classification. In ICLR 2021.
> > > >
> > > > [13] Invariant Models for Causal Transfer Learning. In JMLR 2018.
> > > >
> > > > [14] Disentangled Federated Learning for Tackling Attributes Skew via Invariant Aggregation and Diversity Transferring. In ICML 2022.

---

> > > > ### Comment · Reviewer_LhTp · 2023-11-21
> > > > **Thanks for the clarification**
> > > >
> > > > I want to thank the authors for the clarification and the update on the paper. It is definitely more clear to me now.

---

> > > > > ### Author Response · Authors · 2023-11-21
> > > > > **Thanks for your swift reply and insightful comments**
> > > > >
> > > > > Dear Reviewer #LhTp,
> > > > >
> > > > > We're grateful for your quick feedback during this busy period. We remain open and ready to delve into any more questions or suggestions you might have. Your constructive comments have significantly helped us to refine our work.
> > > > >
> > > > > Best regards and thanks,
> > > > >
> > > > > Authors of #4656

---

### Official Review · Reviewer_9583 · 2023-11-01

**Soundness:** 3 good
**Presentation:** 3 good
**Contribution:** 3 good
**Rating:** 8
**Confidence:** 2

**Summary:**

The paper proposes FedImpro to construct similar features across clients for better federated training. The paper theoretically shows that the "generalization contribution" of local training is bounded by the conditional Wasserstein distances.

**Strengths:**

1. The idea of generalization contribution in FL sounds novel.
2. Experimental performances of FedImpro look superior.

**Weaknesses:**

The idea of having a lower-level and a higher-level neural network in FL is not new, i.e. the feature extraction network idea. I don't see many comparisons to these previous work in the experimental section.

**Questions:**

I don't really follow how "generalization contribution is bounded by conditional Wasserstein distance" leads to the proposed FedImpro algorithm. I think the connection and logic flow can be made more clear.

---

> ### Author Response · Authors · 2023-11-16
> **Response to Reviewer #9583 [1/2]**
>
> We would like to thank the reviewer for taking the time to review. We appreciate that you find that our idea of generalization contribution in FL sounds novel, and experimental performances of FedImpro look superior. According to your insightful and valuable comments, we provide detailed feedback below and add them into our main text or appendix in the revision. We hope these modifications and our revision can address your concerns and improve our work.
>
>
> **Q1** Novelty concerns:
> > The idea of having a lower-level and a higher-level neural network in FL is not new, i.e. the feature extraction network idea. I don't see many comparisons to these previous work in the experimental section.
>
> ***Ans for Q1):*** Thanks for the insightful comments and constructed suggestions. We firstly provide more discussions on differences between FedImpro and these related works. Then we provide an experimental comparison between the most related works that also aim to enhance the global model performance with utilizing features.
> - There are some related works that focus on feature extraction or spliting lower and higher level model. We have discussed them in related works. We summarize the differences between FedImpro and these works and in Table 3-1 as below. Works [1,2] share raw data with noise. FedDF [3] finetunes on the aggregated model via the knowledge distillation relying on shared publica data. Methods [4,5,6,7] enhance model performance by data generation or sharing data statistics. CCVR [8] is a post-hoc method, transmiting the logit statistics of data samples to calibrate the last layer of Federated models. Different from them. [9] aims to defend poisoning attacks. [10-15] directly share hidden features to enhance personal or global federated learning. Different from these methods, FedImpro focuses on **the training process**, constructing **similar feature distribution**, **reducing gradient dissimilarity** of the high-level layers, **without any extra data**, and **sharing estimator parameters instead of raw hidden features**.
>
> Table 3-1
> |                                              | Shared Thing                     | Low-level Model | Objective                  |
> |----------------------------------------------|----------------------------------|-----------------|----------------------------|
> | [1,2] | Raw Data With Noise               | Shared          | Others                     |
> | [3] | Extra Data               | Shared          | Others                     |
> | [4,5] | Params. of Data Generator        | Shared          | Global Model Performance   |
> | [6,7] | STAT. of raw Data                | Shared          | Global Model Performance   |
> |  [8] | STAT. of Logis, Label Distribution | Shared        | Global Model Performance   |
> | [9] | Hidden Features                  | Shared          | Defend Poisoning Attack    |
> | [10,11] | logits                           | Private         | Personalized FL            |
> | [12,13]| Hidden Features                  | Private         | Personalized FL            |
> | [14,15] | Hidden Features                  | Shared          | Accelerate Training        |
> | **FedImpro**                                 | Params. of Estimated Feat. Distribution | Shared   | Global Model Performance   |
>
> - Specifically, we conducted experiments comparing our method with feature sharing methods, such as CCVR [8] and FedDF [3]. As CCVR and our methods do not need extra public training data. FedDF utilizes the noise dataset to conduct knowledge distillation following their original paper [3]. The communication round of our method is reduced to 100 for fair comparison. The results are reported in Table 3-2 as shown below, where our method outperforms existing feature sharing methods. FedDF severely relies on the extra meaningful data to enhance the model performance.
>
> **Table 3-2**
> |               |  CCVR | FedDF with noise data  | Ours   |
> |:-:|:-:|:-:|:-:|
> | Test Accuracy (%)| 62.68 | 38.6 | **68.29** |
>
> We have incorporated corresponding discussion and experimental comparisons in our revision to enhance the quality of our work.

---

> ### Author Response · Authors · 2023-11-16
> **Response to Reviewer #9583 [2/2]**
>
> **Q2** Understanding of the theoretical part:
> > I don't really follow how "generalization contribution is bounded by conditional Wasserstein distance" leads to the proposed FedImpro algorithm. I think the connection and logic flow can be made more clear.
>
> ***Ans for Q2):***
> Thank you for the suggestion aimed at making our work clear. Besides the previous response, we will add the following explanations to our revision. We will provide more detailed explanations if the reviewer finds some points **hard to follow or improper**.
>
> - Our work involves two theorems, i.e., Theorem 4.1 and Theorem 4.2, where Theorem 4.1 motivates the proposed method and Theorem 4.2 indicates another advantage of the proposed method.
> - Theorem 4.1 is built upon two definitions, i.e., Definition 3.1 and Definition 4.1. Here, Definition 3.1 measures the model's generalizability on a data distribution from the margin theory, and Definition 4.1 is the conditional Wasserstein distance for two distributions.
> - Theorem 4.1 shows that promoting the generalization performance requires constructing similar conditional distributions. This motivates our method, i.e., aiming at making all client models trained on similar conditional distributions to obtain higher generalization performance.
> - In our work, inspired by Theorem 4.1, we regard latent features as the data discussed in Theorem 4.1. Accordingly, we can construct similar conditional distributions for the latent features and train models using these similar conditional distributions.
> - Theorem 4.2 is built upon Definition 4.2, where gradient dissimilarity in FL is quantitatively measured by Definition 4.2.
> - Theorem 4.2 shows that our method can reduce gradient dissimilarity to benefit model convergence.
> - For advanced architectures, e.g., a well-trained GFlowNet [3], reducing the distance in the feature space can induce the distance in the data space. However, the property could be hard to maintain for other deep networks, e.g., ResNet.
>
> We have updated the above explainations in appendix for easier understanding. Thanks again for the valuable suggestions, which make our work easier to follow.
>
>
> > ***Reference***
> >
> > [1] Compressive learning with privacy guarantees. In Information and Inference: A Journal of the IMA. 2021.
> >
> > [2] Data synthesis via differentially private markov random fields. In VLDB. 2021.
> >
> > [3] Ensemble Distillation for Robust Model Fusion in Federated Learning. In NeurIPS 2020.
> >
> > [4] G-pate: Scalable differentially private data generator via private aggregation of teacher discriminators. In NeurIPS 2021.
> >
> > [5] Towards fair federated learning with zero-shot data augmentation. In CVPR 2021.
> >
> > [6] Fedmix: Approximation of mixup under mean augmented federated learning. In ICLR 2021.
> >
> > [7] Xor mixup: Privacy-preserving data augmentation for one-shot federated learning. In Arxiv 2020.
> >
> > [8] No Fear of Heterogeneity: Classifier Calibration for Federated Learning with Non-IID Data. In NeurIPS 2021.
> >
> > [9] Cronus: Robust and heteroge- neous collaborative learning with black-box knowledge transfer. In Arxiv 2019.
> >
> > [10] Fedmd: Heterogenous federated learning via model distillation. In Arxiv 2019.
> >
> > [11] Distributed distillation for on-device learning. In NeurIPS 2020.
> >
> > [12] Group knowledge transfer: Federated learning of large cnns at the edge.. In NeurIPS 2020.
> >
> > [13] Think locally, act globally: Federated learning with local and global representations. In Arxiv 2020.
> >
> > [14] Splitfed: When federated learning meets split learning. In Arxiv 2020.
> >
> > [15] Locfedmix-sl: Localize, federate, and mix for improved scalability, convergence, and latency in split learning. In Proceedings of the ACM Web Conference 2022.
> >
> > [16] GFlowNet Foundations. In JMLR 2023.

---

> ### Author Response · Authors · 2023-11-18
> **Welcome for more discussions**
>
> Dear reviewer #9583,
>
> Thanks for your valuable time in reviewing and insightful comments. Following your comments, we have tried our best to provide responses and revise our paper. Here is a **summary of our response** for your convenience:
> - (1) **Novelty concern**: We have discussed many related works that also utilize features to enhance FL from different perspectives. We provide a comprehensive comparison and summarize them in Table 3-1 in response. We also conduct experiments to compare FedImpro with two works that directly utilize features to enhance FL, including CCVR and FedDF. The results shown in Table 3-2 in response illustrate that FedImpro outperforms existing related works.
> - (2) **Illustration the theoretical part and algorithm**: Thanks for your valuable suggestion. We have added an outline to illustrate our theory and how it motivate the algorithm in response and revision.
>
> We humbly hope our repsonse has addressed your concerns. If you have any additional concerns or comments that we may have missed in our responses, we would be most grateful for any further feedback from you to help us further enhance our work.
>
>
> Best regards
>
> Authors of #4656

---

> ### Author Response · Authors · 2023-11-20
> **Window for responsing and draft updating is closing**
>
> Dear Reviewer #9583,
>
> Thanks a lot for your efforts in reviewing paper and your constructive comments. We understand you’re busy. Thanks very much for your valuable comments. We sincerely understand you’re busy. But as the window for responsing and paper revision is closing, would you mind checking our response ([a brief summary](https://openreview.net/forum?id=giU9fYGTND&noteId=RKzZEDFYJt), and [details](https://openreview.net/forum?id=giU9fYGTND&noteId=OVt7G7jsfQ)) and confirm whether you have any further questions? We look forward to answering more questions from you.
>
> Best regards and thanks,
>
> Authors of #4656

---

> > ### Comment · Reviewer_9583 · 2023-11-21
> > **Thanks for the clarification**
> >
> > I thank the authors for providing clarifications on comparison with some related works and how the theoretic findings motivate the proposed method. I am happy to raise my score.

---

> > > ### Author Response · Authors · 2023-11-21
> > > **Thanks for your swift reply and raising the score**
> > >
> > > Dear Reviewer #9583,
> > >
> > > We're grateful for your quick feedback during this busy period. We deeply appreciate your consideration in raising the score. We remain open and ready to delve into any more questions or suggestions you might have. Your constructive comments has significantly contributed to the refinement of our work.
> > >
> > > Best regards and thanks,
> > >
> > > Authors of #4656

---

### Official Review · Reviewer_AtvR · 2023-11-01

**Soundness:** 3 good
**Presentation:** 2 fair
**Contribution:** 2 fair
**Rating:** 6
**Confidence:** 3

**Summary:**

This paper proposed FedImpro, a framework which constructs similar conditional distributions for local training. And FedImpro decouples the model into high-level and low-level parts and trains the high-level part on reconstructed feature distributions, causing promoted generalization contribution and alleviated gradient dissimilarity of FL.

**Strengths:**

1. This paper has a good level of writing and it is easy to follow. The idea is easy to follow and understand.
2. This paper combine split training with feature sharing to improve the generalization of the model.

**Weaknesses:**

1. I notice that the author ignore a very related and state-of-art baesline FedDyn [1], could the author conduct comparion experiments with FedDyn?

2. The timecomsuming for training the model increases for FedImpro. Could the author list the cpu-time cost comparion experiments to reach the target accuracy?


[1] Acar, Durmus Alp Emre, et al. "Federated learning based on dynamic regularization." *arXiv preprint arXiv:2111.04263* (2021).

**Questions:**

See Weaknesses.

---

> ### Author Response · Authors · 2023-11-16
> **Response to Reviewer #AtvR [1/2]**
>
> We would like to thank the reviewer for taking the time to review. We appreciate that you find that our paper is well written and easy to follow, the idea is easy to follow and understand, the generalization of the model is improved. In light of your insightful comments, we offer comprehensive responses below. We also add them into our main text or appendix in the revision. We hope that these responses and the revision have effectively resolved your concerns and enhanced the overall excellence of our work.
>
>
> **Q1** Baseline issues:
> > I notice that the author ignore a very related and state-of-art baesline FedDyn [1], could the author conduct comparion experiments with FedDyn?
>
> ***Ans for Q1):*** Following your valuable suggestions, we compare our method with FedDyn [1] as below.
> - FedDyn proposes to dynamically update the risk objective to ensure the device optima is asymptotically consistent. Different from FedDyn, FedImpro enhances convergence and model performance by constructing more similar feature distribution and reducing gradient heterogeneity, which is orthogonal to FedDyn or other FL algorithms discussed in related works.
> - We conduct experiment comparisons with FedDyn for all settings. The results of FedDyn together with FedAvg and FedImpro are given in Table 2-1 as below. The results show that **FedImpro achieves better final accuracy than FedAvg and FedDyn**. We have added these new results in Table 1 of the main text. FedDyn is observed with much poorer performance when data heterogeneity is severe, which is also observed in [4].
>
>
> **Table 2-1**
> | Dataset     | a     | E   | M    | FedAvg | FedDyn | FedImpro       |
> |-------------|-------|-----|------|----------------|----------------|----------------|
> | **CIFAR-10** |  0.1   | 1   | 10   | 83.65±2.03     |84.73±0.81 | **88.45±0.43** |
> |             |   0.05  | 1   | 10   | 75.36±1.92     |77.21±1.25| **81.75±1.03** |
> |             |  0.1   | 5   | 10   | 85.69±0.57     |86.59±0.39| **88.10±0.20** |
> |             |    0.1   | 1   | 100  | 73.42±1.19     | 75.29±0.95| **77.56±1.02** |
> |  **FMNIST** | 0.1   | 1   | 10   | 88.67±0.34     |89.01±0.25| **90.83±0.19** |
> |              | 0.05  | 1   | 10   | 82.73±0.98     |83.20±1.19| **86.42±0.71** |
> |              | 0.1   | 5   | 10   | 87.6±0.52      |88.50±0.52| **89.87±0.21** |
> |              | 0.1   | 1   | 100  | 90.12±0.19     | 90.57±0.21 | **90.98±0.15** |
> | **SVHN**     | 0.1   | 1   | 10   | 88.20±1.21 | 90.82±1.09 | **92.37±0.82**|
> |           | 0.05   | 1   | 10   | 80.67±1.92  | 84.12±1.28 | **90.25±0.69**|
> |        | 0.1   | 5   | 10   | 86.32±1.19 | 89.92±0.50 | **91.58±0.72**|
> |        | 0.1   | 1   | 100   | 92.42±0.21 |92.82±0.13 | **93.42±0.15**|
> | **CIFAR-100**     | 0.1   | 1   | 10   | 69.38±1.02 | 69.59±0.52 | **70.28±0.33**|
> |           | 0.05   | 1   | 10   | 63.80±1.29  |64.90±0.69 | **66.60±0.91**|
> |        | 0.1   | 5   | 10   | 68.39±1.02 |68.52±0.41 | **68.79±0.52**|
> |        | 0.1   | 1   | 100   | 53.22±1.20 | 54.82±0.81 | **56.07±0.75**|

---

> ### Author Response · Authors · 2023-11-16
> **Response to Reviewer #AtvR [2/2]**
>
> **Q2** Efficiency issue:
> > The timecomsuming for training the model increases for FedImpro. Could the author list the cpu-time cost comparion experiments to reach the target accuracy?
>
>
> ***Ans for Q2):*** Thanks for your insightful comments that are strongly related to the practical deployments. We provided the detailed feedbacks as below.
>
> - **Computational Overhead.** FedImpro requires little more computing time of FedAvg in simulation. Almost all current experiments are usually simulated in FL community [5,6,7]. When setting *local epoch = 1*, *communication rounds = 1000*, the simulation time of the FedAvg is around 8h 52m 12s, while FedImpro consumes around 10h 32m 32s. According to the Table 2 in the main text, FedImpro achieves the 82% acc in similar cpu time with FedAvg. Interestingly, when *local epoch = 5*, the simulation time of FedAvg increases to 20h 10m 25s. The convergence speed of FedImpro with *local epoch = 1* is better than FedAvg with *local epoch = 5*. **Note that in this simulation, the communication does not happen. However, the communication overhead dominates the system costs in the real-world deployment**.
>
> - **Communication Overhead.** Communiation round is a more important real-world metric compared to cpu time in simulation. In real-world federated learning, the bandwidth with the Internet (around $1\sim10$ MB/s) is very low comparing to the cluster environemnt (around $10\sim 1000$ GB/s). Taking ResNet18 with 48 MB as example, each communication with 10 clients would consume $48\sim 480$s, 1000 rounds requires $48000\sim 480000$s ($15\sim 150$ hours). And some other factors like the communication latency and unstability will further increase the communication time. **Thus, the large costs mainly exist in communication overhead instead of the computing time [2,3]. The extra communication size of the estimation parameters in FedImpro is only 4KB, which is almost zero compared with the model size 48MB**. As shown in Table 2 in original text, FedImpro achieve mush faster convergence than other algorithms.
>
>
>
>
> > ***Reference***
> >
> > [1] Acar, Durmus Alp Emre, et al. "Federated learning based on dynamic regularization." ICLR (2021).
> >
> > [2] Communication-efficient learning of deep networks from decentralized data. In AISTATS 2017.
> >
> > [3] Advances and Open Problems in Federated Learning. In Foundations and Trends® in Machine Learning 2021.
> >
> > [4] Sun, Yan, et al. "Fedspeed: Larger local interval, less communication round, and higher generalization accuracy." ICLR (2023).
> >
> > [5] FedML: A Research Library and Benchmark for Federated Machine Learning. In Arxiv 2020.
> >
> > [6] FedScale: Benchmarking Model and System Performance of Federated Learning at Scale. In ICML 2022.
> >
> > [7] A Survey on Federated Learning Systems: Vision, Hype and Reality for Data Privacy and Protection. In TKDE 2023.

---

> ### Author Response · Authors · 2023-11-18
> **Welcome for more discussions**
>
> Dear reviewer #AtvR,
>
> Thanks for your valuable time in reviewing and insightful comments. Following your comments, we have tried our best to provide responses and revise our paper. Here is a **summary of our response** for your convenience:
>
> - (1) **Baseline issues**: Following your constructive comments, we have discussed FedDyn to highlight our novelty. And we further conduct comprehensive experiments of FedDyn. Results are shown in the given Table 2-1 in responses, which is also merged in Table 1 in main text to enhance the quality of our paper. These new results show that **FedImpro achieves better final accuracy than previous baselines and FedDyn**.
> - (2) **Efficiency issue**: We have provided a detailed analysis on the extra time costs of FedImpro. The running time consists of two parts, computation and communication overheads. For the computation overhead, FedImpro requires some extra backward and forward time, **while also showing similar or faster convergence than FedAvg**. However, communication overhead is significantly larger than computation overhead, thus dominating the running time in the real-world FL. For example, when deploying FL with ResNet-18, 10 clients, the computation time might be less than 10 hours, while the communication time may be over than 100 hours. FedImpro shows faster convergence speed with communication round than other baselines according to Table 2 in main text. Thus, **FedImpro can significantly accelerate real-world deployments of FL**.
>
> We humbly hope our repsonse has addressed your concerns. If you have any additional concerns or comments that we may have missed in our responses, we would be most grateful for any further feedback from you to help us further enhance our work.
>
>
> Best regards
>
> Authors of #4656

---

> ### Author Response · Authors · 2023-11-20
> **Window for responsing and draft updating is closing**
>
> Dear Reviewer #AtvR,
>
> Thanks a lot for your time in reviewing and reading our response and the revision. Thanks very much for your valuable comments. We sincerely understand you’re busy. But as the window for responsing and paper revision is closing, would you mind checking our response ([a brief summary](https://openreview.net/forum?id=giU9fYGTND&noteId=Bywr4aah3L), and [details](https://openreview.net/forum?id=giU9fYGTND&noteId=1vYQQ1zcqV)) and confirm whether you have any further questions? We look forward to answering more questions from you.
>
> Best regards and thanks,
>
> Authors of #4656

---

> ### Author Response · Authors · 2023-11-21
> **Window for discussion and revision is closing**
>
> Dear Reviewer #AtvR,
>
> Thanks a lot for your time in reviewing and insightful comments, according to which we have carefully revised the paper to answer the questions. We sincerely understand you’re busy. But since the discussion due is approaching, would you mind checking the response and revision to confirm where you have any further questions?
>
> We are looking forward to your reply and happy to answer your further questions.
>
> Best regards
>
> Authors of #4656

---

> ### Comment · Reviewer_AtvR · 2023-11-21
>
> Thanks for you reply. I will raise my score to 6.

---

> > ### Author Response · Authors · 2023-11-21
> > **Thanks for your swift reply and raising the score**
> >
> > Dear Reviewer #AtvR,
> >
> > Thank you for your prompt response during this busy period. We deeply appreciate your consideration in adjusting the score. If you have any additional questions, we are also very glad to answer. Your insightful comments have been instrumental in improving the quality of our work.
> >
> > Best regards and thanks,
> >
> > Authors of #4656

---

### Official Review · Reviewer_Kzyf · 2023-11-01

**Soundness:** 3 good
**Presentation:** 3 good
**Contribution:** 2 fair
**Rating:** 8
**Confidence:** 3

**Summary:**

The paper addresses the issue of client drift arising in heterogeneous federated settings. As shown by recent literature, local models drifting away from the convergence points of the global model lead to slower and unstable convergence. This work first shows how the generalization contribution is bounded by the conditional Wasserstein distance between clients’ local data distributions. To reduce the client drift, FedImpro decouples local training into two stages: the lower part of the model is trained on the local data, while the higher part is also trained on shared reconstructed features. Empirical results on FL benchmark datasets show the efficacy of FedImpro in terms of final performance, convergence speed, and reduced weights divergence.

**Strengths:**

- The paper addresses a very relevant issue for the FL community, i.e. limiting the negative effects of the client drift in heterogeneous settings.
- The paper is well written and easy to follow
- Very detailed discussion of related works
- Theoretical claims supported by proofs
- Extensive empirical analysis. FedImpro is compared with some state-of-the-art approaches in terms of final performance, convergence speed, weight divergence. Interesting ablation study on the depth of gradient decoupling. Results compared under different levels of client participation, local training epochs, data heterogeneity, and settings (cross-device vs cross-silo FL).
- Limitations regarding communication and privacy concerns are explicitly addressed. It is shown how FedImpro does not lead to privacy leakage in the gradient inversion attack.
- All details for reproducing the experiments are provided.

**Weaknesses:**

- My main concern regards the feasibility of deploying FedImpro in real-world contexts. FedImpro notably increases both the number of communications between clients and server, and the message size. The paper points out how the global distribution can be estimated using methods which impact the communication network less, but that does not eliminate the need for additional communication.
- Some relevant related works are not discussed: ETF [1], SphereFed [2], FedSpeed [3].
- FedImpro is compared with relatively old baselines (published 3 years ago). More recent and relevant baselines are for instance FedDyn [4], FedSpeed, ETF. Also, I believe CCVR and SphereFed are highly correlated with FedImpro and it is relevant to understand how they behave w.r.t. FedImpro in terms of client drift reduction, final performance, communication costs, even if they are only applied at the end of training.
- From Fig. 2, the actual impact of FedImpro on reducing the weight divergence appears very limited.

**Minor weaknesses and typos:**
- Lack of NLP datasets, e.g. Shakespeare and StackOverflow, or large scale datasets, e.g. Landmarks-User-160k.
- Typo (?) page 7: I believe the explanation of cross-device or cross-silo setting should be switched, i.e. M=100 for cross-device FL and M=10 for cross-silo FL.

**References**
[1] Li, Zexi, et al. "No Fear of Classifier Biases: Neural Collapse Inspired Federated Learning with Synthetic and Fixed Classifier." ICCV (2023).
[2] Dong, Xin, et al. "Spherefed: Hyperspherical federated learning." European Conference on Computer Vision. Cham: Springer Nature Switzerland, 2022.
[3] Sun, Yan, et al. "Fedspeed: Larger local interval, less communication round, and higher generalization accuracy." ICLR (2023).
[4] Acar, Durmus Alp Emre, et al. "Federated learning based on dynamic regularization." ICLR (2021).

**Questions:**

1. How does FedImpro compare with more recent FL methods addressing data heterogeneity? Please refer to Weaknesses for examples on related works.
2. How much bigger is FedImpro's impact on the communication both in terms of number of communications between clients and server, and message size w.r.t. FedAvg and the other introduced FL baselines?
3. How does FedImpro compare with CCVR and SphereFed w.r.t. centralized performances?

---

> ### Author Response · Authors · 2023-11-16
> **Response to Reviewer #Kzyf [1/2]**
>
> We sincerely thank the reviewer for taking the time to review. We appreciate that you find that we address a relevant issue for the FL community, our paper is well written and easy to follow, detailed discussion of related works, theoretical proof, extensive empirical analysis, interesting ablation study and high reproducibility. According to your valuable comments, we provide detailed feedback below and also add them into our main text or appendix in the revision. We hope that these changes have addressed your concerns and improved the overall quality of our work.
>
>
> **Q1** Feasibility issues:
> > FedImpro notably increases both **(a)** the number of communications between clients and server, and **(b)** the message size. The paper points out how the global distribution can be estimated using methods which impact the communication network less, but that does not eliminate the need for additional communication.
>
> ***Ans for Q1):*** Thanks for your insightful comments for the real-world deployments. We provide following analysis of the extra communication overhead. Specifically, the communication time in real-world can be characterized by the alpha-beta model [1,2]: $T_{comm}=\alpha + \beta M$, where $\alpha$ is the latency per message (or say each communication), $\beta$ the inverse of the communication speed, and $M$ the message size. In FedImpro, the estimated parameters are communicated along with the model parameters. Thus
> - **(a)** **the number of communications is not increased**, and the communication time from $\alpha$ will not increase;
> - **(b)** The $M$ increased as $M_{model} + M_{estimator}$. Taking ResNet-20 as an example, the size of estimated parameters is equal to doubled (mean and variance) feature size (not increased with batch size) as 4KB (ResNet-20), which is greatly less than the model size around 48 MB. Therefore, **the additional communication cost is almost zero**.
>
>
> **Q2** Related work issues:
> > Some relevant related works are not discussed: ETF [3], SphereFed [4], FedSpeed [5].
>
> ***Ans for Q2):*** Thanks for the provided references from the reviewer, which are definitely related to our work. We have added discussions into the related works in revision. The following detailed discussions are listed in revised appendix due to the limited space of the main text.
> - FedETF [3] proposed a synthetic and fixed ETF classified to resolve the classifier delemma, which is orthogonal to FedImpro.
> - SphereFed [4] proposed constraining learned representations of data points to be a unit hypersphere shared by clients. Specifically, in SphereFed, the classifier is fiexed with weights spanning the unit hypersphere, and calibrated by a mean squared loss. Besides, SphereFed discovers that the non-overlapped feature distributions for the same class lead to weaker consistency of the local learning targets from another perspective. FedImpro alleviate this problem by estimating and sharing similar features.
> - FedSpeed [5] utilized a prox-correction term on local updates to reduce the biases introduced by the prox-term, as a necessary regularizer to maintain the strong local consistency. FedSpeed further merges the vanilla stochastic gradient with a perturbation computed from an extra gradient ascent step in the neighborhood, which can be seen as reducing gradient heterogeneity from another perspective.

---

> ### Author Response · Authors · 2023-11-16
> **Response to Reviewer #Kzyf [2/2]**
>
> **Q3** Baseline issues:
> > More recent and relevant baselines are for instance FedDyn [6], FedSpeed, ETF. Also, I believe CCVR and SphereFed are highly correlated with FedImpro and it is relevant to understand how they behave w.r.t. FedImpro in terms of client drift reduction.
>
> ***Ans for Q3):*** Following your valuable suggestions, we compare our method with SphereFed[4], FedDyn[6], CCVR [7], and FedSpeed [5]. The results are given in Table 1-1 as below. As original experiments in these works do not utilize the same FL setting, we report their test accuracy of CIFAR-10/100 dataset and the according settings in original papers. To make the comparison fair, we choose the same or the harder settings of their methods and report their original results. The larger dirichlet parameter $\alpha$ means more severe data heterogeneity. The higher communication round means training with longer time. We can see that **our method can outperform these methods with the same setting or even the more difficult settings for us**.
>
> **Table 1-1**
> | Method | Dataset | Dirichlet $\alpha$ |  Client Number | Communication Round | Test ACC. |
> |:-:|:-:|:-:|:-:|:-:|:-:|
> | FedDyn[6] | CIFAR-10 | 0.3 | 100 | 1400  | 77.33 |
> | FedSpeed[5] | CIFAR-10 |0.3 | 100 | 1400 | 82.68 |
> | FedImpro | CIFAR-10 |0.3 | 100 | 1400 | **85.14** |
> | FedImpro | CIFAR-10 |0.1 | 100 | 1400 | **82.76** |
> |--------|--------|---------|---------|--------|------|
> | CCVR[7] | CIFAR-10 | 0.1 | 10 | 100 |  62.68 |
> | FedImpro | CIFAR-10 | 0.1 | 10 | 100 | **68.29** |
> |--------|--------|---------|---------|--------|------|
> | SphereFed [4] | CIFAR-100 | 0.1 | 10 | 100 |  69.19 |
> | FedImpro | CIFAR-100 | 0.1 | 10 | 100 | **70.28** |
>
> Those previously baselines in the main paper are commonly compared in the recent FL works. Thanks again, we have added these results and analysis to our revision.
>
>
>
> **Q4** Weight divergence issues:
> > From Fig. 2, the actual impact of FedImpro on reducing the weight divergence appears very limited.
>
> ***Ans for Q4):*** Thanks for your careful reading of our paper and insightful comments.
> - During the previous trainig stage, the weight divergence redution is limited, because the **low-level model and the feature estimation are still unstable**. After about 500 communication rounds, FedImpro shows lower weight divergence than others.
> -  As Fig. 2 (a), Fig. 4, 5, 6 and 7 (in Appendix) show,  although in the early training stage the weight divergence reduction is limited, **the training convergence can  benefit from FedImpro**. Future work can think about how to fasten convergence of low-level models.
>
>
> **Q5** NLP dataset:
> > Lack of NLP datasets, e.g. Shakespeare and StackOverflow, or large scale datasets, e.g. Landmarks-User-160k.
>
> ***Ans for Q5):*** Thanks for indicating this problem. We conducted experiments with image datasets, but omiting the NLP datasets in this work. We will explore new experiments using datasets with different modalities including language, graph, tabular data, and so on in future works.
>
>
> **Q6** Typos:
> > Typo in page 7.
>
> ***Ans for Q6):*** Thanks for your careful reviewing and the kind comments. Yes, the cross-silo FL and cross-device FL should be switched. We have modified the typo and update the text in the revision.
>
>
>
>
>
> > ***Reference***
> >
> > [1] Optimization of collective communication operations in MPICH. In IJHPCA 2005.
> >
> > [2] MG-WFBP: Efficient data communication for distributed synchronous SGD algorithms. In INFOCOM. 2019
> >
> > [3] Li, Zexi, et al. "No Fear of Classifier Biases: Neural Collapse Inspired Federated Learning with Synthetic and Fixed Classifier." ICCV (2023).
> >
> > [4] Dong, Xin, et al. "Spherefed: Hyperspherical federated learning." European Conference on Computer Vision. Cham: Springer Nature Switzerland, 2022.
> >
> > [5] Sun, Yan, et al. "Fedspeed: Larger local interval, less communication round, and higher generalization accuracy." ICLR (2023).
> >
> > [6] Acar, Durmus Alp Emre, et al. "Federated learning based on dynamic regularization." ICLR (2021).
> >
> > [7] No Fear of Heterogeneity: Classifier Calibration for Federated Learning with Non-IID Data. NeurIPS 2021.

---

> ### Author Response · Authors · 2023-11-18
> **Welcome for more discussions**
>
> Dear reviewer #Kzyf,
>
> Thanks for your valuable time in reviewing and constructive comments, according to which we have tried our best to answer the questions and carefully revise the paper. Here is a **summary of our response** for your convenience:
>
> - (1) **Feasibility issues**: We provide analysis of the extra communication overhead, consisting of the number of communications and the additional communication size. With sending the estimated parameters along with model parameters per communication, **the number of communications is not increased**. And the size of estimated parameters is greatly smaller than the model size, thus **the additional communication cost is almost zero**.
> - (2) **Related work issues**: Following your constructive comments, we have discussed related works including FedETF, SphereFed, and FedSpeed to highlight our novelty. And we also add these discussions into our revision to enhance our work.
> - (3) **Baseline issues**: Following your valuable suggestions, we have compared our method with SphereFed, FedDyn, CCVR, and FedSpeed. The given results show that **our method can outperform these methods with the same setting or even the more difficult settings**.
> - (4) **Weight divergence issues**: We provide more explanations to illustrate why weight divergence reduction is small during early trainig stage. While, it is reduced a lot during late stage and benefit FL training.
>
> We humbly hope our repsonse has addressed your concerns. If you have any additional concerns or comments that we may have missed in our responses, we would be most grateful for any further feedback from you to help us further enhance our work.
>
>
> Best regards
>
> Authors of #4656

---

> ### Author Response · Authors · 2023-11-20
> **Window for responsing and draft updating is closing**
>
> Dear Reviewer #Kzyf,
>
> Thanks very much for your time and valuable comments. We understand you're busy. But as the window for responsing and paper revision is closing, would you mind checking our response ([a brief summary](https://openreview.net/forum?id=giU9fYGTND&noteId=jym8mCSO5R), and [details](https://openreview.net/forum?id=giU9fYGTND&noteId=osgtdzhtxo)) and confirm whether you have any further questions? We are very glad to provide answers and revision to your further questions.
>
> Best regards and thanks,
>
> Authors of #4656

---

> ### Comment · Reviewer_Kzyf · 2023-11-20
> **Response to authors' rebuttal**
>
> I thank the authors for addressing my concerns and taking my suggestions into account.
> I believe their clarifications and additional comparison with more recent state-of-the-art approaches further validates the quality of the submitted work. I'm happy to raise my score.

---

> > ### Author Response · Authors · 2023-11-21
> > **Thanks for your swift reply and raising the score**
> >
> > Dear Reviewer #Kzyf,
> >
> > Thanks for your swift reply despite such a busy period. We sincerely appreciate that you can raise the score. If you have any further questions or comments, we are very glad to discuss more with you. Your valuable comments have greatly helped us in enhancing our work.
> >
> > Best regards and thanks,
> >
> > Authors of #4656

---

### Meta-Review · Area_Chair_evrs · 2023-12-09

**Metareview:**

Reviewers are universally positive about this paper; the authors responded satisfactorily addressing most of the concerns. This is recommended for acceptance.

**Justification For Why Not Higher Score:**

The paper is not being proposed for spotlight due to a lack of excitement in reviewers; one concrete reason for that is the high-level ideas to mitigate client drift are present in some form in the literature (such as federated representation learning, fine-tuning).

**Justification For Why Not Lower Score:**

Nonetheless the contribution is solid enough that the ICLR audience will benefit from knowing the algorithms.

---

### Decision · Program_Chairs · 2024-01-16

Accept (poster)